# Tankyrase disrupts metabolic homeostasis and promotes tumorigenesis by inhibiting LKB1-AMPK signalling

Nan Li[1]*, Yifan Wang [1,2,3], Shinya Neri[1], Yuanli Zhen[4], Lon Wolf R. Fong[2,5], Yawei Qiao[1], Xu Li[1,6], Zhen Chen[1], Clifford Stephan [7], Weiye Deng [3,8], Rui Ye[1,2], Wen Jiang [3,8], Shuxing Zhang [5], Yonghao Yu[4], Mien-Chie Hung [2,9], Junjie Chen[1,2]* & Steven H. Lin[1,2,8]*

The LKB1/AMPK pathway plays a major role in cellular homeostasis and tumor suppression. Down-regulation of LKB1/AMPK occurs in several human cancers and has been implicated in metabolic diseases. However, the precise upstream regulation of LKB1-AMPK pathway is largely unknown. Here, we report that AMPK activation by LKB1 is regulated by tankyrases. Tankyrases interact with and ribosylate LKB1, promoting its K63-linked ubiquitination by an E3 ligase RNF146, which blocks LKB1/STRAD/MO25 complex formation and LKB1 activation. LKB1 activation by tankyrase inhibitors induces AMPK activation and suppresses tumor-igenesis. Similarly, the tankyrase inhibitor G007-LK effectively regulates liver metabolism and glycemic control in diabetic mice in a LKB1-dependent manner. In patients with lung cancer, tankyrase levels negatively correlate with p-AMPK levels and poor survival. Taken together, these findings suggest that tankyrase and RNF146 are major up-stream regulators of LKB1-AMPK pathway and provide another focus for cancer and metabolic disease therapies.

[1] Department of Experimental Radiation Oncology, The University of Texas MD Anderson Cancer Center, Houston, TX 77030, USA. [2] The University of Texas Graduate School of Biomedical Sciences Houston, Houston, TX 77030, USA. [3] Department of Radiation Oncology, University of Texas Southwestern Medical Center, Dallas, TX 75219, USA. [4] Department of Biochemistry, University of Texas Southwestern Medical Center, Dallas, TX 75390, USA. [5] Department of Experimental Therapeutics, The University of Texas MD Anderson Cancer Center, Houston, TX 77030, USA. [6] School of Life Science, Westlake University, Hangzhou 310024, China. [7] Center for Translational Cancer Research, The Institute of Biosciences and Technology at Texas A&M University, Houston, TX 77030, USA. [8] Department of Radiation Oncology, The University of Texas MD Anderson Cancer Center, Houston, TX 77030, USA. [9] Department of Molecular and Cellular Oncology, The University of Texas MD Anderson Cancer Center, Houston, TX 77030, USA. *email: nli4@mdanderson.org; jchen8@mdanderson.org; SHLin@mdanderson.org

iver kinase B1 (LKB1, also called STK11) is a serine/threonine kinase and is a key regulator of cellular energy homeostasis. The gene encoding LKB1 was discovered as a tumor-suppressor gene linked to Peutz-Jeghers syndrome (PJS)[1]. Patients with PJS are tumor-prone, and germline mutations in LKB1 are common in patients with sporadic cancers, including lung adenocarcinoma[2,3] and cervical carcinoma[4]. Lkb1-deficient mice die at mid-gestation, suggesting that LKB1 has an essential role in mouse development[5]. Lkb1-heterozygous mice develop gastrointestinal polyps and hepatocellular carcinoma[6], supporting the tumor-suppressor role of LKB1.

As a protein kinase, LKB1 is the major upstream kinase of the energy sensor AMPK (AMP-activated protein kinase) under energy stress[7]. LKB1 phosphorylates AMPK at threonine 172 to activate AMPK[8], activated AMPK then phosphorylates downstream targets involved in fatty acid synthesis, glycolysis, and protein synthesis. The AMPK downstream targets include ACC1 in fatty acid synthesis[9], Raptor in protein synthesis[10] and others. Thus, the LKB1-AMPK pathway has central roles in cell metabolism, survival, and proliferation in response to nutrient and energy levels[11].

LKB1 exhibits weak catalytic activity in vivo and in vitro, and its activation is predominantly stimulated by the formation of LKB1 complex with two proteins, STRAD and MO25[12,13]. The STRAD (STE20-related pseudokinase) binds to the kinase domain of LKB1 and activates its kinase activity[12], and MO25 enhances the effects of STRAD on LKB1[14].

In this study, we identify tankyrase inhibitors as potent activators of AMPK. We demonstrate that tankyrase functions as a key regulator of AMPK activation through ribosylating LKB1. We further find that the ribosylation of LKB1 by tankyrase is specifically recognized by the E3 ligase RNF146, which led to ubiquitination of LKB1 and disruption of the LKB1/STRAD/MO25 complex. On the other hand, a tankyrase inhibitor show both anti-tumor and antidiabetic effects via activating the AMPK pathway. We describe a molecular mechanism by which the tankyrase-RNF146 axis regulates the LKB1-AMPK pathway, and uncover tankyrase as a drug target for lung cancer therapy and glycemic control.

## Results

**Tankyrase inhibition leads to AMPK activation**. As a critical sensor of cellular response to energy stress, AMPK plays important roles in various metabolic processes[7]. To identify potential regulators of AMPK, we screened the small-molecule Informer-Set[15] by using a phosphor-AMPK ELISA assay as indications of AMPK activation. First, we chose metformin and AICAR as positive controls[16,17], and we tested the activation rates of pT172-AMPK in different cell lines (293A, H1299, H2087, MCF7, H358, and U2OS). We found that U2OS cells were most sensitive to metformin/AICAR induced AMPK activation (data not shown) and we choose U2OS cells for the assay (Supplementary Data 1). Notably, in addition to metformin/ AICAR, we identified JW55, JW-55, and CHIR-99021 as putative AMPK activators (Fig. 1a). The GSK-3 inhibitor CHIR-99021 was previously shown to activate AMPK[18] and both JW55 and JW-55 are inhibitors of tankyrase. Therefore, tankyrase inhibitors might be regulator of AMPK activation.

Tankyrase1/PARP5a and tankyrase2/PARP5b are members of the poly(ADP-ribose) polymerase (PARP) family. The enzymatic activities of these tankyrases are similar to those of PARP1 in catalyzing the poly-ADP-ribosylation of substrate proteins[19,20]. Normally, tankyrases recognize substrates through a six-amino-acid RxxxxG motif[21]. Most tankyrase-mediated PARylation is recognized by a specific PAR-binding E3 ligase RNF146, which

leads to substrate ubiquitination[22]. Tankyrases were originally discovered as regulator of telomere elongation by ribosylating TRF1[19], and were recently identified to regulate diverse functions, including in regulation of wnt pathway[23], PI3K-Akt signaling[24], Hippo-Yap pathway[25], and pexophagy[26].

To validate the role of tankyrase inhibitors in AMPK activation, we treated six cell lines with JW55 and another tankyrase inhibitor, G007-LK, which was reported to be active in xenograft models[27]. Both JW55 and G007-LK led to AMPK activation without changes in LKB1/AMPK levels (Supplementary Fig. 1a). To confirm this observation, we tested the effects of different tankyrase inhibitors (JW55, WIKI4, XAV939, and G007-LK)[23,27–29] on AMPK activation. Similar to metformin/ AICAR, we found all tankyrase inhibitors increased the phosphorylation of AMPK and the AMPK substrate ACC (acetyl-CoA carboxylase) in HEK293A cells; however, PARP1/2 inhibitor olaparib[30] had no effect (Fig. 1b). Similar observations were noted for the mRNA expression of PDK1, FAS (fatty acid synthase) and SREBP-1c, three AMPK downstream glycolytic genes (Supplementary Fig. 1b)[31–33]. The tankyrase inhibitor XAV939 induced AMPK activation in a dose-dependent manner (Supplementary Fig. 1c), and tankyrase inhibitors regulated AMPK phosphorylation without changing the AMP/ATP ratio (Supplementary Fig. 1d). Collectively, these results indicate that tankyrase inhibitors are positive regulators of AMPK activation.

As treatment with tankyrase inhibitors led to stabilization of tankyrases substrates Axin[23], and Axin was implicated in AMPK activation[34], we next investigated whether the AMPK activation induced by tankyrase inhibitors was through Axin. We found that tankyrase inhibitors drastically stimulated AMPK activation in Axin-depleted cells (Supplementary Fig. 1e, f), indicating that the effect of tankyrase inhibitors on AMPK activation depends predominantly on tankyrases and to a lesser extent Axin stability.

Next we found that overexpression of TNKS1/2 suppressed AMPK activation and that this suppression was blocked by the tankyrase inhibition (Supplementary Fig. 1g). We further found that the wild-type, but not the catalytic-inactive mutant of TNKS1 (TNKS1-H1184/E1291A or TNKS1-PD), led to the downregulation of phosphor-AMPK (Supplementary Fig. 1h), indicating that catalytic activity of tankyrases is required for AMPK regulation.

As AMPK has critical roles in metabolic processes including lipid metabolism and glycolysis, and activation of AMPK leads to decreased cell proliferation in normal conditions but increased cell survival in response to energy stress[7], we tested the effect of tankyrases on these processes. We observed that only TNKS1/2 double-knockdown resulted in AMPK activation (Fig. 1c), significant suppression of AMPK downstream glycolic genes expression (Fig. 1d), lipid droplet formation (Supplementary Fig. 1i), decreased cell proliferation (Supplementary Fig. 1j), and reduction of cell death in response to energy stress (Supplementary Fig. 1k). Moreover, reintroduction of TNKS1, but not TNKS1-PD could rescue TNKS1/2 double depletion triggered AMPK activation and energy stress-induced cell death (Supplementary Fig. 1l, m). These findings implicate tankyrases in AMPK activation.

**Tankyrases associate with and PARylate LKB1**. To explore the mechanism underlying tankyrase inhibitor–induced AMPK activation, we examined the interactions between tankyrases and key components of the LKB1-AMPK pathway. Results showed that both TNKS1/2 interacted with LKB1 but not AMPKa (Supplementary Fig. 2a), and these associations were confirmed by an endogenous immunoprecipitation (Fig. 1e). Two putative tankyrase-binding motifs were found in LKB1 but not in AMPK

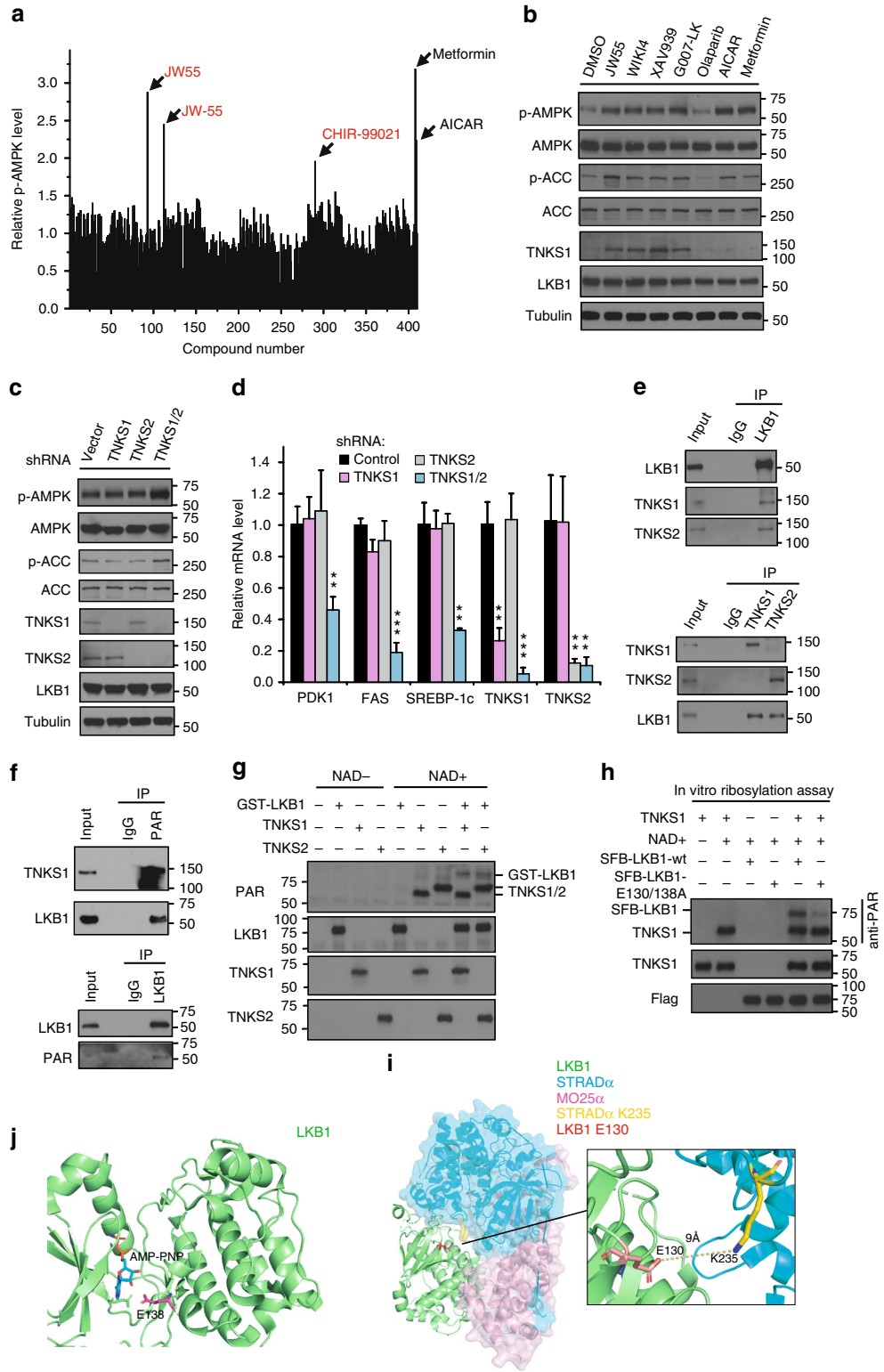

components (Supplementary Fig. 2b), and these motifs were highly conserved (Supplementary Fig. 2c). Interestingly, structural analyses of the LKB1/TNKS complex (the crystal structures for LKB1 and the TNKS ANK repeat2 were taken from the RCSB Protein Data Bank) revealed that G47-Y49 and R86-A93 from LKB1 are located proximal to the interface of the LKB1/ TNKS–ANK2 complex (Supplementary Fig. 2d, e), indicating that these residues might influence the LKB1/tankyrase interaction. To test that possibility, we created point mutations and found

that the mutated version (LKB1-R42/G47/R86/G91A) could not interact with TNKS1/2 (Supplementary Fig. 2f), confirming the importance of these tankyrase-binding motifs on LKB1 for tankyrase–LKB1 interactions. Together, these findings suggest that LKB1 is a binding partner of tankyrases.

Next, we sought to determine whether LKB1 could be ribosylated by tankyrases. Indeed, LKB1 ribosylation was observed in vivo (Fig. 1f). We also performed an in vitro ribosylation assay (Fig. 1g) and found that TNKS1/2 both

**Fig. 1** Tankyrases Regulate AMPK Activation through ribosylating LKB1. **a** Identification of the tankyrase inhibitor JW55 as an AMPK activator. A drug screen for regulators of AMPK was done by treating U2OS cells with a set of 406 drugs (the small-molecule Informer-Set contains 481 drugs, but we can only get 406 drugs from commercial sources) and metformin/AICAR. Both JW55 and JW-55 were the same compounds from two different commercial sources. **b** Tankyrase inhibitors specifically induce AMPK activation. HEK293A cells were treated with indicated inhibitors and proteins were detected by western blotting. **c** Double-knockdown of TNKS1/2 induces AMPK activation. HEK293A cells were transduced with the indicated shRNA and subjected to western blotting. **d** Depletion of TNKS1/2 suppresses expression of AMPK downstream glycolytic genes. Relative expression levels of PDK1, FAS SREBP-1c, and TNKS1/2 were detected by qPCR. **e** Interactions between endogenous LKB1 and TNKS1. Immunoprecipitation/western blotting were performed with the indicated antibodies. **f** LKB1 is ribosylated in vivo. Lysates from HEK293A cells were subjected to immunoprecipitation/western blotting assays. **g** LKB1 is ribosylated by TNKS1 in vitro. In vitro ribosylation was assessed by using recombinant TNKS1/LKB1 (E.coli purified GST-LKB1 and enzymatic active form of TNKS1, amino acids 1000–1328 with GST-tag). **h** E130 and E138 on LKB1 are key residues for TNKS1-induced ribosylation of LKB1. In vitro ribosylation was assessed by using the indicated proteins. **i** Structure view of LKB1/STRAD/MO25 complex. The LKB1-E130 residue is located close to the interface between LKB1 and STRAD. **j** Structure view of LKB1. The LKB1-E138 residue is located close to the ATP-binding pocket of LKB1. Statistical significance was determined by a two-tailed, unpaired Student's *t*-test. *$P < 0.05$; **$P < 0.01$; ***$P < 0.001$

ribosylated themself and LKB1 directly. The tankyrase inhibitors XAV939 and G007-LK abolished the ribosylation of TNKS1 and LKB1 efficiently (Supplementary Fig. 2g); moreover, only the wild-type, but not the enzymatic-inactive mutant (TNKS1-PD) of TNKS1, ribosylated itself and LKB1 (Supplementary Fig. 2h). We further confirmed that inhibition or knockdown of tankyrases suppressed LKB1 ribosylation levels in vivo (Supplementary Fig. 2i), and only tankyrase inhibitors but not PARP1/2 inhibitors specifically suppressed ribosylation of LKB1 (Supplementary Fig. 2j), indicating that LKB1 can be ribosylated by tankyrases.

Next, we used in vitro ribosylation assays and mass spectrometry to analyze ribosylation sites on LKB1. The ADP-ribosylation sites were determined using the hydroxylamine chemistry, which converts an ADP-ribosylated D/E residue into a hydroxamic acid derivative[35]. We identified three specific peptides with five putative ribosylation sites on LKB1 (Supplementary Fig. 2k, l. Supplementary Data 2a, b, c). To further characterize the modification sites, we generated different D-to-A or E-to-A mutations in the putative residues (D30, E33/92/130/138) and found that both E130 and E138 are critical for LKB1 ribosylation (Supplementary Fig. 2m, n). Further analysis indicated that both E130/138 residues are conserved (Supplementary Fig. 2o), and the LKB1-E130/138 A mutant abolished the ribosylation of LKB1 (Fig. 1h). Structural analyses of LKB1 revealed that LKB1 E130 is located near the interface of the LKB1/STRAD complex[13] (Fig. 1i), and that LKB1 E138 is located near the ATP-binding pocket of LKB1[13] (as indicated by the ATP analog AMP-PNP) (Fig. 1j). These findings indicate that the E130/138 ribosylation sites may affect LKB1 function.

**Regulation of AMPK activation by tankyrases depends on LKB1 status.** As LKB1 was ribosylated by tankyrase directly, we assessed whether regulation of AMPK activation depends on LKB1. We generated stably expressed LKB1 in the LKB1-null lung cancer cell line A549 (Fig. 2a), and a stable knockout of LKB1 in the LKB1 wild-type mouse lung adenocarcinoma cell line LKR13 (Fig. 2b). We observed that the tankyrase inhibition led to AMPK activation only in LKB1-positive cells (Fig. 2a, b). Moreover, tankyrase inhibition suppressed AMPK downstream genes expression (Supplementary Fig. 3a, b), cell proliferation (Fig. 2c), and colony formation (Fig. 2d), all to a considerably greater degree in LKB1-positive cells, indicating that the functions of tankyrase in AMPK pathway depend on LKB1.

To determine the effect of ribosylation on LKB1 function, we generated stable clones expressing wild-type LKB1, a non-ribosylation mutant of LKB1 (LKB1-E130/138 A) or a tankyrase-binding mutant of LKB1 (LKB1-R42/138AG47/R86/G91A) in the LKB1-deficient HeLa cell line. We found the LKB1-E130/138 A and LKB1-R42/138AG47/R86/G91A mutants showed greater phosphorylation of AMPK, but did not respond

to tankyrase inhibitor–mediated regulation of AMPK phosphorylation (Fig. 2e) and were resistant to tankyrase inhibitor–induced decreases in cell proliferation and colony formation (Fig. 2f, g). Moreover, the LKB1-E130/138 A mutant showed more resistant to tankyrase inhibitor–induced decreases in lipid droplets formation (Supplementary Fig. 3c), a greater response to metformin-induced AMPK phosphorylation (Supplementary Fig. 3d) and a greater protective effect on energy stress–induced cell death (Supplementary Fig. 3e). These results suggest that tankyrase-induced LKB1 ribosylation at the E130/E138 sites is critical for tankyrase-dependent LKB1 regulation.

We also tested the effect of tankyrase-inhibitor treatment on xenograft tumors. In the LKR13 lung tumor model, LKR13-LKB1-knockout tumors grew much more aggressively than LKR13 tumors (Fig. 2h–j). However, tankyrase inhibitor G007-LK had a substantial tumor-suppressive effect primarily on LKR13 tumors and much less so on LKR13-LKB1-ko tumors (Fig. 2h–k). In addition, the anti-tumor effect of G007-LK was observed in Hela-wtLKB1 tumors, but not in Hela or Hela-mtLKB1 tumors (Fig. 2l). Collectively, these findings demonstrate that the anti-tumor effect of G007-LK depends on LKB1.

**RNF146 inhibits AMPK activation by mediating K63-Ubiquitination of LKB1.** Reasoning that tankyrases regulate the LKB1-AMPK pathway by modifying LKB1, and realizing that the E3 ligase RNF146 is critical for the regulation of tankyrase substrates[22,24,25], we examined whether RNF146 is involved in the regulation of the LKB1-AMPK pathway. Indeed, loss of RNF146 resulted in an increase in AMPK/ACC phosphorylation (Fig. 3a), a significant decrease in AMPK downstream gene expression (Fig. 3b), cell proliferation (Supplementary Fig. 4a) lactate production (Supplementary Fig. 4c), lipid droplet formation (Supplementary Fig. 4d), and a protective effect on metabolic stress–induced cell death (Supplementary Fig. 4b). All indicate that the AMPK pathway was activated after RNF146 knockdown.

We further noted that overexpression of wild-type but not the mutant RNF146 led to suppression of AMPK (Fig. 3c). Notably, treatment with tankyrase inhibitors increased the phosphorylation of AMPK/ACC and reversed the RNF146-induced AMPK suppression (Fig. 3c). Given that the WWE domain is required for the recognition of tankyrase substrates, and the RING domain is critical for the E3 ligase activity of RNF146[22], these findings suggest that the recognition of LKB1, and the E3 ligase activity of RNF146 all contribute to RNF146-mediated AMPK regulation.

Next we checked the association between RNF146 and LKB1 (Fig. 3d), and found that XAV939, which abolished LKB1 ribosylation, disrupted the RNF146/LKB1 interaction (Fig. 3e). We also found that only wild-type, but not two mutants RNF146 promoted LKB1 ubiquitination (Fig. 3f). Notably, because knockdown of RNF146 led to AMPK activation but did not

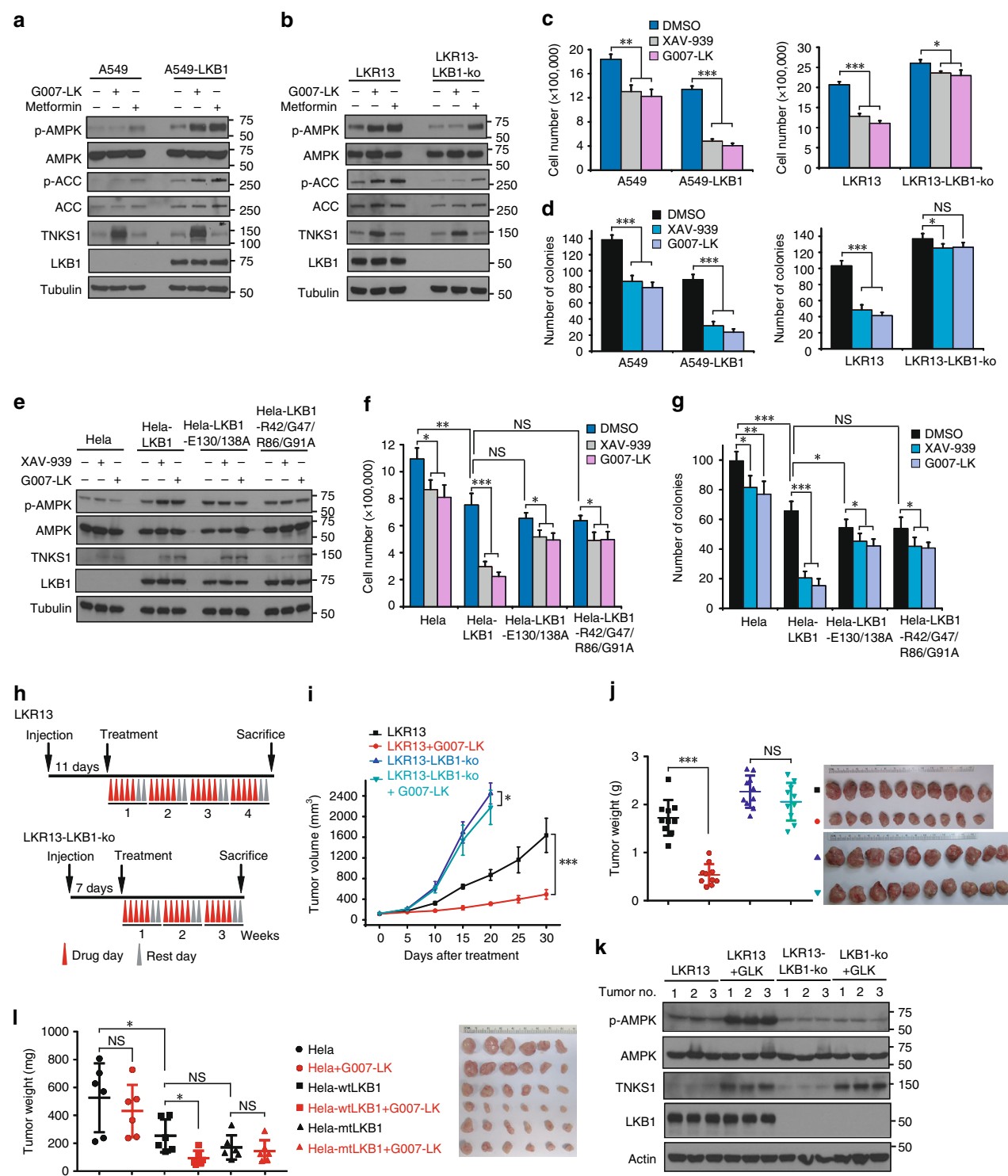

affect LKB1 protein levels (Fig. 3a), we reasoned that RNF146 could regulate the LKB1-AMPK pathway through a nondegradative mechanism. As K48-linked ubiquitination normally triggers proteasome degradation but K63-linked ubiquitination is the signal for nondegradative functions[36], we considered the possibility that RNF146 may regulate AMPK activation by K63-linked ubiquitination. We used K48R and K63R-ubiquitin for an in vivo ubiquitination assay and found that K63R, but not K48R-ubiquitin, blocked RNF146-mediated LKB1 ubiquitination (Fig. 3g). We also found that RNF146 only induced incorporation of K63-ubiquitin into LKB1 (Supplementary Fig. 4e) and that

deletion of RNF146 only decreased K63-ubiquitin–induced LKB1 ubiquitination (Supplementary Fig. 4f). Furthermore, loss of RNF146 led to significant reduction of K63-linked ubiquitination of endogenous LKB1 (Fig. 3h). Given that LKB1 was ribosylated by tankyrase and recognized by RNF146, we examined the response of wild-type and the LKB1-E130/138 A mutant to RNF146 overexpression. We observed that expression of RNF146 led to suppression of AMPK but only in the LKB1 wild-type and not in the LKB1-E130/138 A mutant cells (Supplementary Fig. 4g). We also found that the E130/138 A mutant could not interact with RNF146 (Supplementary Fig. 4i), and RNF146 and

**Fig. 2** LKB1 is Essential for Tankyrase-Mediated AMPK Regulation. **a** Expression of LKB1 in LKB1-deficient A549 cells restores G007-LK-induced AMPK activation. **b** G007-LK induced AMPK activation is impaired in LKB1-knockout cells. LKR13 and LKR13-LKB1-knockout cells were treated with the indicated inhibitors were subjected to western blotting. **c**, **d** Tankyrase inhibitor–induced suppression of cell growth (**c**) and colony formation (**d**) depends on LKB1. Cell proliferation and colony formation were measured as described above; data are presented as means ± SD ($n = 3$ independent experiments). **e** The LKB1-E130/138 A mutant is resistant to tankyrase inhibitor–induced AMPK activation. HeLa cells stably expressing wild-type LKB1 or the LKB1-E130/138 A mutant were treated with XAV939 or G007-LK and the cell lysates were subjected to western blotting. **f**, **g** Tankyrase inhibitors have much less of an effect on suppression of cell growth (**f**) and colony formation (**g**) in HeLa cells expressing the LKB1-E130/138 A mutant. Cell proliferation and colony formation were measured. **h** Schematic display of the LKR13 and LKR13-LKB1-knockout mouse xenograft experimental design. LKR13 ($0.5 \times 10^6$ cells) and LKR13-LKB1-ko ($0.25 \times 10^6$ cells) were implanted by injection. **i**, **j** Tumor volume (**i**) and weight (**j**) of mice from different treatment groups; data are presented as means ± SD ($n = 10$ mice). **k** Tumor proteins from the indicated groups of mice were extracted, and lysates were examined by western blotting with the indicated antibodies. **l** Tumor weight of mice with subcutaneous injection of $2 \times 10^6$ indicated HeLa cells (mtLKB1 indicates LKB1-E130/138 A mutant) with vehicle or G007-LK were presented ($n = 6$ mice, mean = ± SD). Statistical significance was determined by a two-tailed, unpaired Student's $t$-test. $*P < 0.05$; $**P < 0.01$; $***P < 0.001$

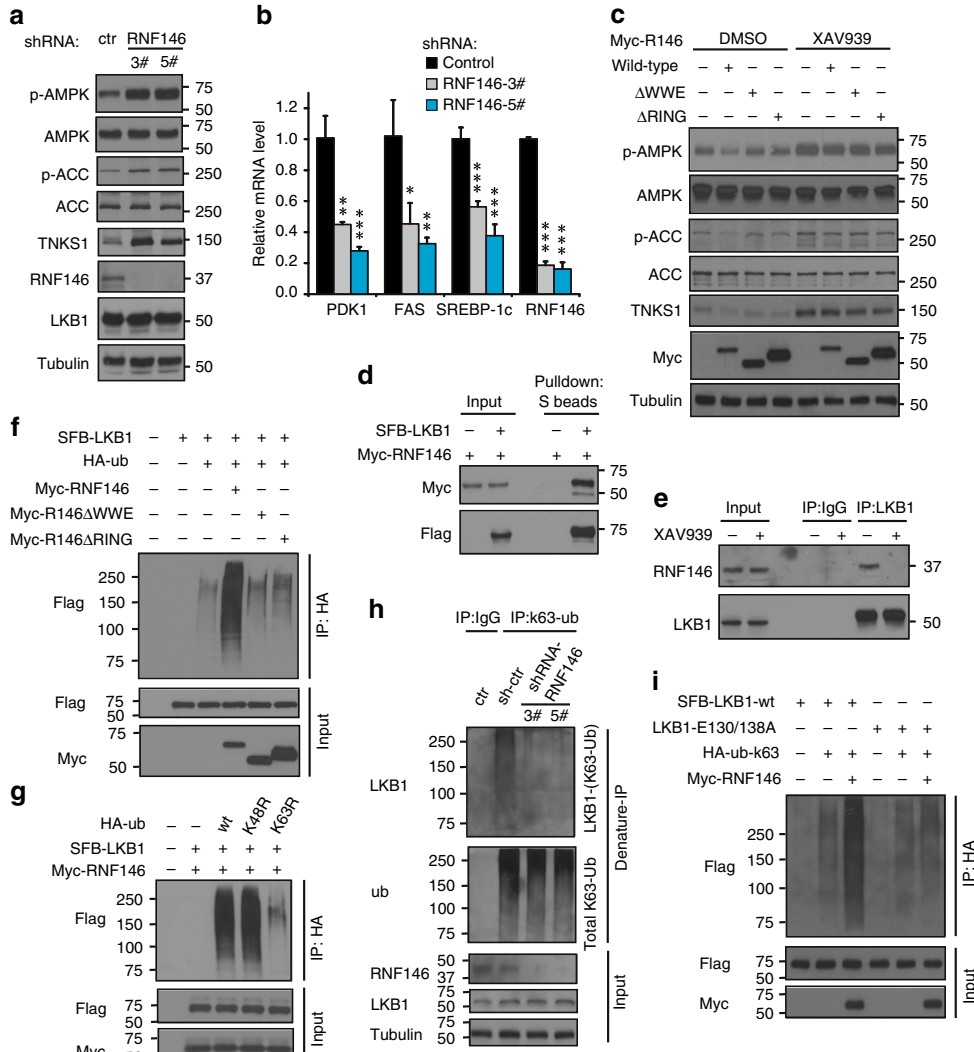

**Fig. 3** RNF146 is Involved in Regulation of the LKB1-AMPK Pathway Through K63-Linked LKB1 Ubiquitination. **a** Knockdown of RNF146 leads to AMPK activation. HEK293A cells were transduced with the indicated shRNA and subjected to western blotting. **b** Loss of RNF146 suppresses expression of AMPK downstream glycolytic genes. The relative expression levels of PDK1, FAS, and SREBP-1c were detected by qPCR. **c**, Both the WWE and RING domains of RNF146 are required for suppression of AMPK phosphorylation. **d** LKB1 interacts with RNF146. Myc-tagged RNF146 was co-expressed with or without SFB-LKB1 and cell lysates were subjected to pull-down assay. **e** Interaction between LKB1 and RNF146 is blocked by tankyrase inhibitor. **f** RNF146 induces LKB1 ubiquitination in vivo. In vivo ubiquitination assay was assessed in 293 T cells transfected with the indicated constructs followed by western blotting. **g** RNF146 promotes LKB1 ubiquitination through K63-linkage. 293 T cells were transfected with the indicated plasmids, and subjected to an in vivo ubiquitination assay. **h** RNF146 depletion diminishes K63-linked LKB1 ubiquitination. K63-linked ubiquitination of endogenous LKB1 was detected by denature-IP using a K63-specific ubiquitin antibody. **i** RNF146 doesn't affect K63-linked ubiquitination of the LKB1-E130/138 A mutant. In vivo ubiquitination assay was performed in 293 T cells transfected with the indicated constructs. Statistical significance was determined by a two-tailed, unpaired Student's $t$-test. $*P < 0.05$; $**P < 0.01$; $***P < 0.001$

tankyrase inhibitor treatment did not affect K63-linked ubiquitination levels of the LKB1-E130/138 A mutant (Fig. 3i and Supplementary Fig. 4h). We conclude that RNF146 is required for K63-linked, non-proteolytic ubiquitination of LKB1, and ribosylation of LKB1 is required for the recognition and ubiquitination of LKB1 by RNF146.

**The Tankyrase-RNF146 axis affects the formation of the LKB1 complex and LKB1 phosphorylation.** Given that formation of the LKB1-STRAD-MO25 complex is critical for LKB1 activation[14], and the fact that one of LKB1 ribosylation site LKB1-E130 is located near the interface of the LKB1/STRAD complex (Fig. 1j), we further investigated the mechanism of tankyrase-RNF146-induced LKB1 regulation. We observed that tankyrase inhibitors led to increased association between the components of the LKB1/STRAD/MO25 complex without affecting their protein levels (Fig. 4a). Furthermore, double-knockdown of TNKS1/2 enhanced the formation of the LKB1/STRAD/MO25 complex (Fig. 4b), whereas rescue of wild-type but not the enzymatic-inactive mutant of TNKS1 in TNKS1/2-knockdown cells reversed the increased formation of the LKB1/STRAD/MO25 complex (Fig. 4c). Consistently, depletion of RNF146 also increased the integrity of the complex (Fig. 4d). Next, we examined the LKB1/STRAD/MO25 complex formation using WT and the E130/138 A mutant of LKB1. We found that the LKB1-E130/138 A mutant formed a stronger complex with STRAD and MO25 (Fig. 4e, f), and tankyrase inhibitor treatment or RNF146 overexpression had no effect on the complex formation between the LKB1-E130/138 A mutant and STRAD/MO25 (Fig. 4e, f). These results implicate tankyrase/RNF146 in LKB1 complex formation (Supplementary Data 3).

Moreover, we also observed an increase of LKB1 phosphorylation at Ser428 in TNKS1/2-depleted (Fig. 4g) or RNF146-depleted cells (Fig. 4h). Phosphorylation of LKB1 at different sites has distinct functions[37,38], especially phosphorylation at Ser428 is important for LKB1-induced AMPK activation[37,39]. Taken together, these results indicate that the tankyrase-RNF146 axis modulates LKB1 function by affecting both complex formation and the phosphorylation of LKB1.

**The tankyrase inhibitor G007-LK improves glycemic control in diabetic mice.** Activation of AMPK in liver can efficiently suppress gluconeogenesis and lower blood glucose levels[40], and the AMPK activator metformin is widely used to reduce blood glucose levels in patients with diabetes[41]. As G007-LK was used in xenograft models[42], we examined whether G007-LK could improve glycemic control in diabetic mice. Treatment with G007-LK significantly reduced blood glucose levels, as did metformin; interestingly, the combination of G007-LK and metformin had a significantly greater effect (Fig. 5a). Moreover, in comparison with vehicle group, all treatment groups had significantly lower plasma insulin levels (Fig. 5b). In addition, the db/db mice had much greater water consumption and urine output than non-diabetic control, and all treatment groups showed drastic reductions in water consumption and urine output (Fig. 5c, d), indicating that G007-LK treatment led to remarkable glycemic control in diabetic mice.

The body weight of the control db/db mice increased only slightly throughout the treatment period (Supplementary Fig. 5a); the metformin-treated mice showed similar, but the mice treated with G007-LK showed slight losses of body weight (Supplementary Fig. 5a, b). Although the reason for this body-weight loss is unknown, it may have been associated with reduced food intake (Supplementary Fig. 5c).

We also examined metabolic changes in the livers of the db/db mice and found that the G007-LK-treated mice had decreased hepatic triglyceride (Fig. 5e) and cholesterol levels (Fig. 5d). We also observed increased LKB1 and AMPK phosphorylation (Fig. 5g) and downregulated expression of AMPK target genes in the livers of G007-LK treated db/db mice (Fig. 5h, i), which was similar to the effect of metformin and consistent with our cell-based data. Taken together, these results suggest that the tankyrase inhibitor G007-LK regulates both liver metabolism and glycemic control in diabetic mice.

**Oncologic relevance of tankyrase expression and AMPK inactivation in lung cancer.** To further investigate the role of tankyrase in lung cancer, we generated doxycycline-inducible expression of TNKS1 in LKR13 cells. Expression of TNKS1 led to AMPK inactivation (Fig. 6a), enhanced cell proliferation and colony formation (Fig. 6b, c), which was suppressed by tankyrase inhibitors (Fig. 6b, c). We also observed a much more aggressive tumor growth in mice injected with LKR13 cells expressing TNKS1 than control cells (Fig. 6d, e), and injection of G007-LK suppressed TNKS1-induced tumor growth (Fig. 6d, e). These data indicated that TNKS1 overexpression promoted tumorigenesis.

To address the clinical relevance of tankyrase expression and AMPK inactivation in lung cancer, we performed a human lung tumor TMAs for pathologic correlation (Supplementary Fig. 6a). We found that both TNKS1 and TNKS2 were overexpressed in human lung carcinoma (Supplementary Fig. 6b), while LKB1 and p-AMPK levels were downregulated (Supplementary Fig. 6b). As expected, a strong positive association between the levels of p-AMPK and LKB1 expression was observed (Supplementary Fig. 6c). Interestingly, p-AMPK level negatively correlated with TNKS1 and TNKS2 expression only in LKB1-positive samples (Fig. 6f) but not in the entire dataset (Supplementary Fig. 6c), indicating the dependence of LKB1 in tankyrase-induced AMPK inactivation. Moreover, Kaplan–Meier analysis of long-term outcomes for overall survival revealed that high expression of TNKS1 was significantly correlated with poorer overall survival in both lung adenocarcinoma and lung squamous cell carcinoma patients (Fig. 6g), as well as for low level of p-AMPK (Fig. 6h) (Supplementary Data 4a, b). These data further support the role of tankyrase-mediated AMPK inactivation and its contribution to lung cancer prognosis.

## Discussion

The findings presented here reveal a role for tankyrases in the regulation of the LKB1-AMPK pathway. By using an unbiased drug screen, we identified tankyrase inhibitors as activating AMPK. We further found that LKB1 is a previously unknown substrate for tankyrases. The PARylation subsequently leads to recognition and K63-linked ubiquitination of LKB1 by the E3 ligase RNF146, and this K63-linked ubiquitination disrupts formation of the LKB1-MO25-STRAD complex and thereby suppresses the activation of LKB1. Notably, tankyrase inhibitors also induced LKB1-AMPK activation, tumor suppression and reduced blood glucose levels in diabetic mice. These findings reveal a role for tankyrases in the regulation of LKB1 and provide a strong rationale for clinical application of tankyrase inhibitors for cancer treatment in LKB1 wild-type tumors and glycemic control of diabetic patients (Supplementary Fig. 7).

As a major upstream kinase of AMPK, LKB1 is known to be regulated in several ways[43]. First, LKB1 activation depends on its assembly into complexes with STRAD and MO25. In this study, we revealed that tankyrases suppressed LKB1 complex assembly and thus inhibited LKB1-AMPK activation (Figs. 1, 4). Another way in which LKB1 is regulated is by post-translational

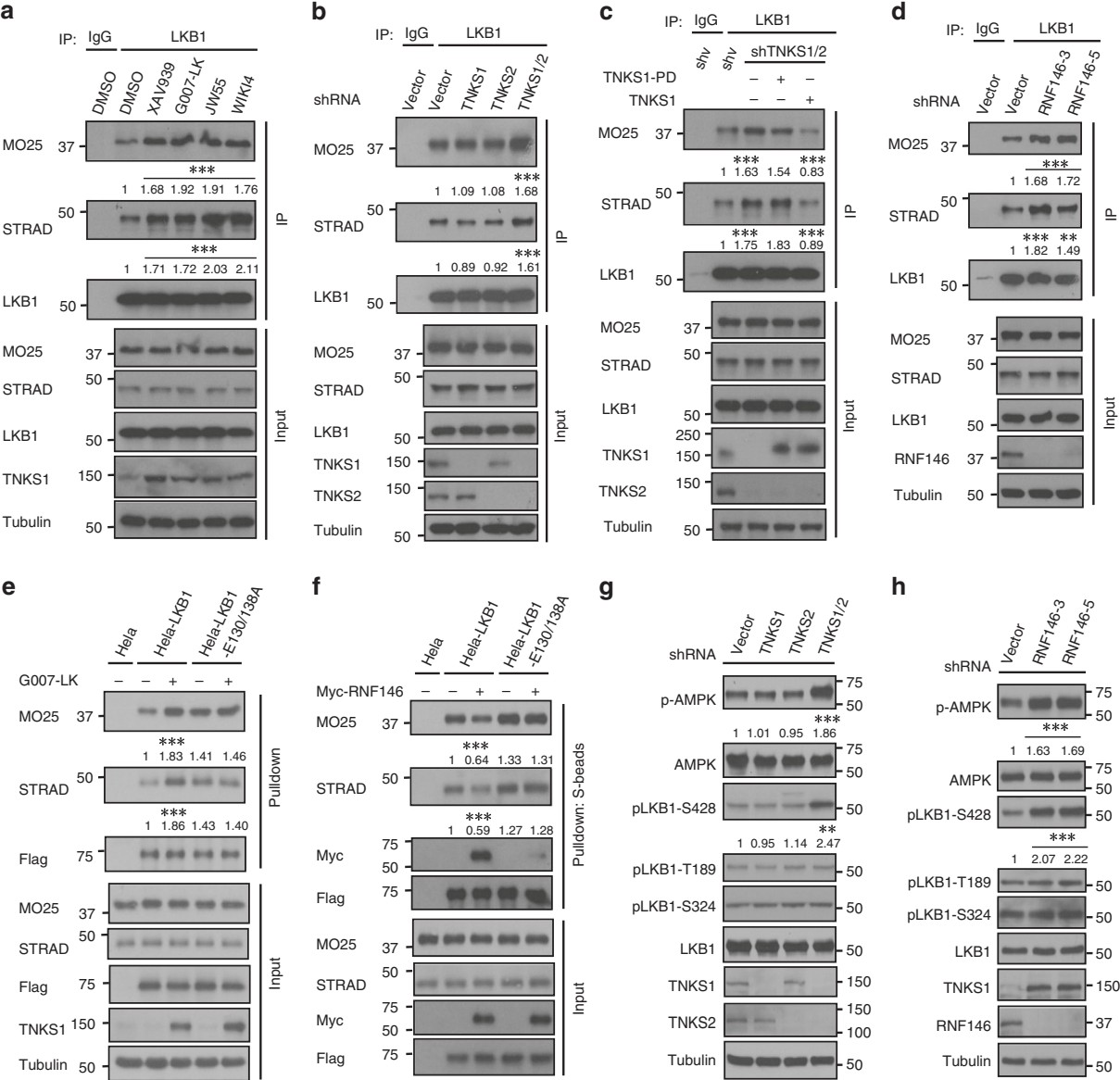

**Fig. 4** Tankyrase-RNF146 axis is involved in regulation of LKB1 complex integrity and LKB1 phosphorylation. **a** Tankyrase inhibitor treatment enhances LKB1/STRAD/MO25 complex formation. HEK293A cells were treated with DMSO or the tankyrase inhibitors JW55. WIKI4, XAV939, or G007-LK, and subjected to immunoprecipitation/western blotting. We use the ImageJ to calculate the band densitometry of western blots, and get the relative ratio by comparing the densitometry of MO25/LKB1 or STRAD/LKB1, and we get three relative ratios from three independent experiments. The relative ratio data was analyzed by the One-way ANOVA test and we used an F-test to compare variances, a P value of <0.05 was considered to indicate statistically significant differences, ** means P-value of <0.01 and *** means P-value of <0.001 (shown in Supplementary Data 4). **b** TNKS1/2 depletion increases LKB1 complex formation. HEK293A cells were transduced with the indicated shRNAs and cell lysates were subjected to immunoprecipitation/western blotting. **c** Expression of TNKS1 but not the TNKS1-PD mutant disrupts LKB1 complex formation. shRNA-resistant inducible TNKS1 or TNKS1-PD were transducted into TNKS1/2-knockdown HEK293A cells, and cell lysates were subjected to immunoprecipitation. **d** Loss of RNF146 promotes LKB1 complex formation. HEK293A cells were transduced with the indicated shRNAs and cell lysates were subjected to immunoprecipitation/western blotting. **e** G007-LK treatment promotes complex formation of WT but not E130/138 A mutant of LKB1 with STRAD/MO25. Indicated cells were treated without or with G007-LK and cell lysates were subjected to pull-down assay. **f** Overexpression of RNF146 inhibits complex formation of WT but not E130/138 A mutant of LKB1 with STRAD/MO25. HeLa cells stable expressed wild-type or the E130/138 A mutant of LKB1 was transfected without or with Myc-RNF146. Cell lysates were subjected to pull-down assay. **g** TNKS1/2 depletion induces LKB1 phosphorylation at Ser428. HEK293A cells were transduced with the indicated shRNAs, and phosphorylation of LKB1 at different sites was assessed with the corresponding antibodies. **h** Knockdown of RNF146 promotes LKB1 phosphorylation. HEK293A cells were transduced with the indicated shRNAs and subjected to western blotting. Statistical significance was determined by a two-tailed, unpaired Student's t-test. *P < 0.05; **P < 0.01; ***P < 0.001

modifications, by being phosphorylated at several residues[12,37], and by being modified by ubiquitination[44,45]. We uncovered previously unreported post-translational modifications of LKB1, one being the tankyrase-mediated PARylation of LKB1 (Fig. 1), which led to K63-linked, non-degradative polyubiquitination of LKB1 by RNF146 (Fig. 3), and the other being tankyrase-RNF146 axis–mediated PARylation and ubiquitination of LKB1, which subsequently affects LKB1 phosphorylation at Ser428 and

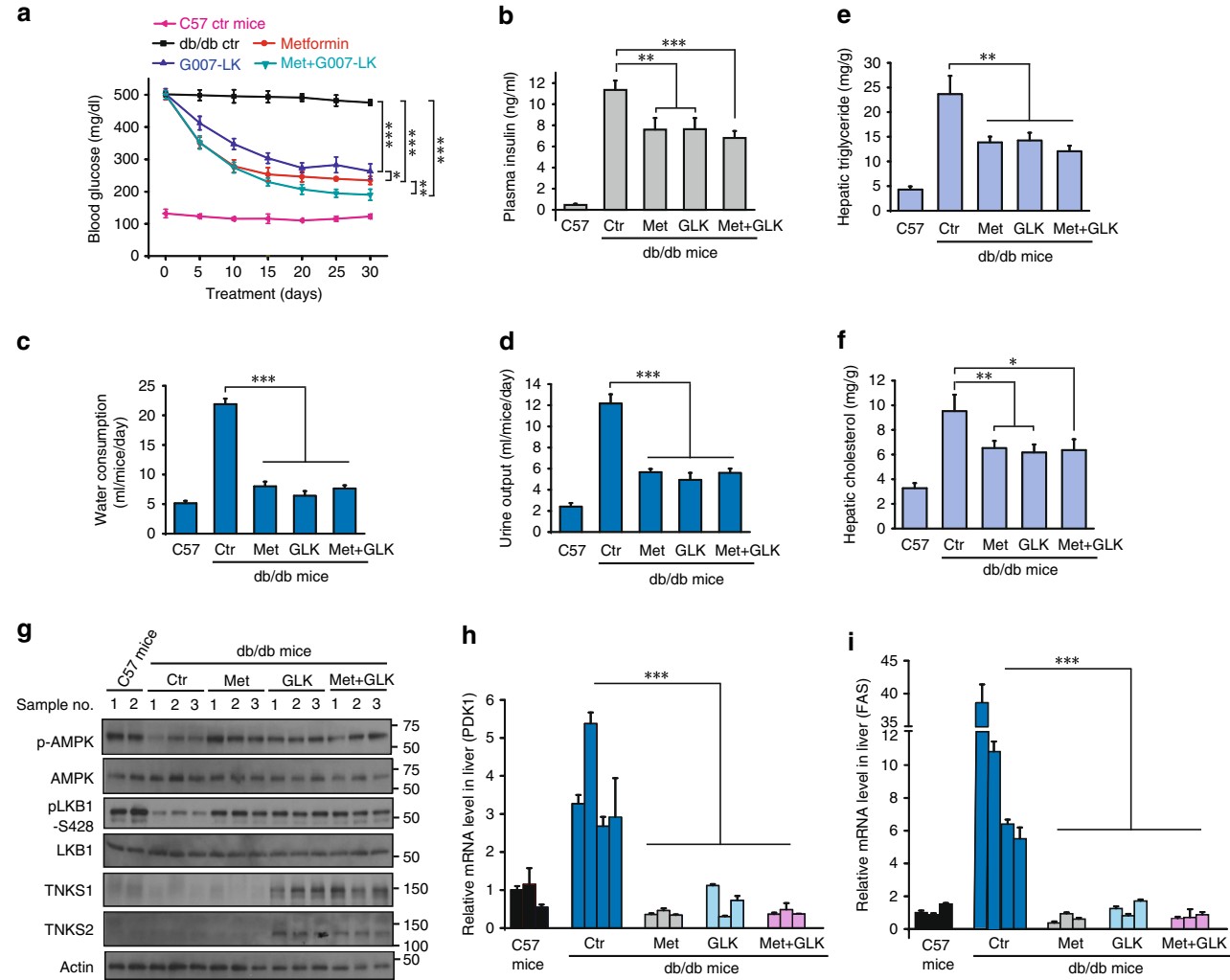

**Fig. 5** Tankyrase Inhibitor G007-LK Improves Glycemic Control in a Diabetic Mouse Model. **a** Tankyrase inhibitor G007-LK lowers blood glucose in db/db mice. The db/db mice were separated into four groups and i.p.-injected with vehicle (PBS), metformin (50 mg/kg), G007-LK (30 mg/kg) or the combination of metformin plus G007-LK daily for 30 days. Blood glucose levels were measured with a glucometer on the indicated days after the mice had been fasted for 18 h. **b** Basal plasma insulin concentrations were measured in each group at day 30. **c, d** Daily water consumption (**c**) and urine output (**d**) were measured after 3 weeks of treatment; data are presented as means ± SD ($n = 5$ independent experiments for water consumption and $n = 3$ for urine output). **e, f** Liver lipids from each group were extracted and hepatic triglyceride (**e**) and cholesterol (**f**) levels were measured. **g** Liver proteins from the indicated groups of db/db mice were extracted, and phosphorylation levels of AMPK and LKB1 were examined by western blotting. **h, i** Liver mRNAs from each group were extracted, and relative expression of PDK1 (**h**) and FAS (**i**) were determined by qPCR; transcript levels were determined relative to mRNA levels and normalized. The results represent the means ± SD ($n = 3$ independent experiments). Statistical significance was determined by a two-tailed, unpaired Student's *t*-test. *$P < 0.05$; **$P < 0.01$; ***$P < 0.001$

assembly of the LKB1-STRAD-MO25 complex (Fig. 4). We believe that this represents an unknown mechanism of LKB1 regulation.

Despite the fact that double-knockout of tankyrases TNKS1 and TNKS2 leads to embryonic lethality in mice[46], TNKS1-deficient mice survive with reduced adiposity[47], and TNKS2-deficient mice survive with growth defects[48]. Interestingly, we observed that treating db/db diabetic mice with the tankyrase inhibitor G007-LK led to reductions in blood glucose and body weight (Fig. 5, Supplementary Fig. 5), findings that agree with the phenotypes of tankyrase-knockout mice. While tankyrase inhibitors have been developed and used to treat colorectal tumors in xenograft models[42], we have expanded the therapeutic indications in models of lung cancer and diabetes (Figs. 5, 6). Although the inhibitory activity of the tankyrase inhibitor G007-LK did not seem to wane for the duration of our preclinical experiments, it is uncertain if this activity will not become refractory with longer term treatments.

Future studies will be needed to explore the durability of this response, as well as the study of other bioavailable drugs such as AZ6102. In summary, we have found an important link of tankyrase to LKB1 regulation, which further advanced our understanding of tankyrase inhibition and its therapeutic potential.

## Methods

**Antibodies and reagents**. For western blotting, rabbit anti-TNKS2 antibody was raised by immunizing rabbits with GST-TNKS2 fusion protein containing amino acids 527–776 of human TNKS2. Anti-RNF146 (ab201212, 1:500 dilution) was obtained from abcam. Anti-HA (H9658, 1:2000 dilution), anti-α-tubulin (T6199, 1:3000 dilution), anti-Flag (M2) (F3165, 1:5000 dilution), and anti-β-actin (A2228, 1:2000 dilution) were purchased from Sigma–Aldrich. Anti-LKB1 (3047 S, 1:1000 dilution), anti-phospho-LKB1 (T189) (3054 S, 1:1000 dilution), anti-phospho-LKB1 (S334) (3055 S, 1:1000 dilution), anti-phospho-LKB1 (S428) (3482 S, 1:1000 dilution), anti-Axin1 (2074 S, 1:1000 dilution), anti-AMPKα (2532 S, 1:1000 dilution), anti-phospho-AMPKα (T172) (2535 S, 1:1000 dilution), anti-ACC (3662 S, 1:1000 dilution), anti-phospho-ACC (S79) (3661 S, 1:1000 dilution), and anti-MO25α (2716 S, 1:3000 dilution) were obtained from Cell Signaling Technology.

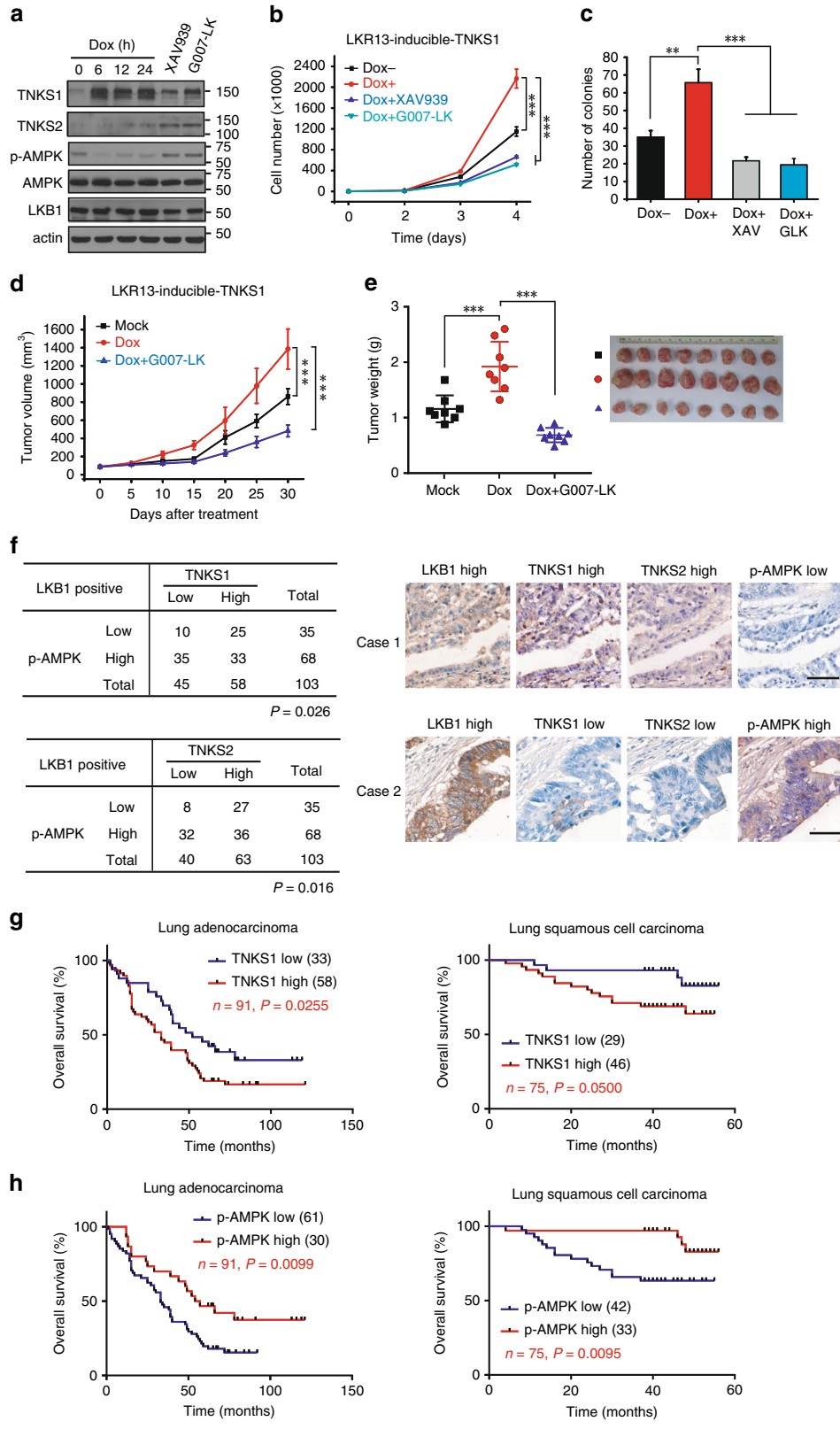

Anti-Myc (sc-40, 1:1000 dilution), anti-LKB1 (sc-374300, 1:1000 dilution), anti-TNKS (sc-8337, 1:1000 dilution), and anti-GST (sc-138, 1:1000 dilution) were purchased from Santa Cruz Biotechnology. Anti-ubiquitin-K63 (05–1313, 1:500 dilution) was obtained from Millipore; anti-STRADα (PA572175, 1:500 dilution) was obtained from Thermo Fisher; and anti-PAR (4335-MC-100, 1:1000 dilution) was obtained from Trevigen.

MG132, metformin, and puromycin were purchased from Sigma–Aldrich. Olaparib, XAV939, JW55, and WIKI4 were from Selleck Chemicals, G007-LK and

AICAR were purchased from ApexBio, and the neutral lipid dye BODIPY 493/503 was obtained from Invitrogen.

**Constructs, shRNAs, gRNAs, and siRNAs.** The TNKS1, TNKS2, and TNKS1-PD ("PARP dead", i.e., the H1184A/E1291A mutant) plasmids were kindly provided by Dr. Susan Smith (New York University); the HA-ubiquitin-K48R/K63R/K48/K63 plasmids were provided by Dr. Li Ma (MD Anderson Cancer Center). The LKB1,

**Fig. 6** Overexpression of tankyrase promotes tumor growth and predicts poor clinical outcome. **a** Inducible expression of TNKS1 leads to AMPK inactivation. LKR13 cells stably expressing inducible TNKS1 were treated without or with doxycycline (dox:100 ng/ml) for the indicated hours, or with XAV939/G007-LK (5 μM) for 12 h followed by western blotting. **b**, **c** Inducible expression of TNKS1 promotes cell proliferation (**b**) and colony formation (**c**). LKR13 cells stably expressing inducible TNKS1 were treated without or with doxycycline (dox:100 ng/ml), or treated in combination with doxycycline plus XAV939/G007-LK, Cell proliferation and colony formation were measured as described above; data are presented as means ± SD (n = 3 independent experiments). **d**, **e** Inducible expression of TNKS1 promotes tumor growth. LKR13-inducible-TNKS1 cells ($0.5 \times 10^6$ cells) were implanted by injection. Tumor volume (**d**) and weight (**e**) of mice from different treatment groups; data are presented as means ± SD ($n = 8$ mice). **f** Quantification and two representative cases of immunohistochemical staining for the correlation between p-AMPK and TNKS1/2 using human lung carcinoma tumor tissue microarray analysis. Statistical significance was determined by Chi-square test. Bar, 50 μm. **g**, **h** Kaplan–Meier overall-survival analysis of TNKS1 (**g**) and p-AMPK (**h**) levels in lung adenocarcinoma and lung squamous cell carcinoma patients detected by IHC staining ($n = 91$ and 75 tissues, respectively, log rank test). Statistical significance was determined by a two-tailed, unpaired Student's $t$-test. *$P < 0.05$; **$P < 0.01$; ***$P < 0.001$

AMPKα, RNF146 constructs were obtained from the Human ORFeome version 5.1 collection. All constructs were sub-cloned into pDONOR201 vector and then into S-protein/Flag/SBP triple-tagged, HA-tagged, and Myc-tagged destination vectors by using Gateway Technology (Invitrogen). The RNF146-delta-WWE and RNF146-delta-RING mutants were generated by deleting amino acids 104–159 or amino acids 36–78, respectively, of the RNF146 coding region. The point mutated version of LKB1 was generated by site-directed mutagenesis and verified by sequencing or synthesized by Integrated DNA Technologies. The TNKS1-inducible plasmid was generated by the gateway cloning technology, the mouse TNKS1 sequence was inserted into a plasmid (pINDUCER21, from Addgene) containing a tetracycline (Doxycycline) response element, and the TNKS1-inducible plasmid was transfected into LKR13 cells. The LKR13-inducible-TNKS1 stable cells were generated by GFP sorting.

The pLKO.1 TNKS1 shRNA was purchased from Sigma (TRCN0000040184). The following pGIPZ lentiviral shRNA sets were from the shRNA and ORFeome Core Facility at The University of Texas MD Anderson Cancer Center, with clone numbers as follows: TNKS2-shRNA (V3LHS_354387); RNF146-shRNA-3# (V2LHS_12922); and RNF146-shRNA-5# (V3LHS_395653). The control shRNA sequence was 5′-TCTCGCTTGGGCGAGAGTAAG-3′. Lentiviral packaging plasmids (pMD2.G and psPAX2) were kindly provided by Dr. Zhou Songyang (Baylor College of Medicine). The sgRNA sequence used for targeting mouse LKB1 was ACTCCGAGACCTTATGCCGC.

The shRNA-resistant expression constructs of TNKS1 and TNKS1-PD were generated by mutating the nucleotide sequence but not changing the amino-acid sequence. The original shRNA targeting sequence for TNKS1 (5′-CGACTCTTAGAGGCATCTAAA-3′) was changed to (5′-AGGTTACTGGAAGCGTCTAAA −3′) and verified by sequencing.

Non-targeting siRNA control (ON-TARGETplus Non-targeting siRNA #1, catalog number D-001810-01-05), TNKS1 siRNA (SMARTpool: ON-TARGETplus TNKS siRNA, catalog number L-004740-00-0005), TNKS2 siRNA (SMARTpool: ON-TARGETplus TNKS2 siRNA, catalog number L-004741-00-0005), and RNF146 siRNA (SMARTpool: ON-TARGETplus RNF146 siRNA, catalog number L-007080-00-0005) were purchased from Dharmacon.

**RNA isolation, reverse transcription, and real-time PCR.** Total mRNA was isolated from cells by using a PARIS Kit (Thermo Fisher). A reverse transcription assay was done with the OneTaq RT-PCR Kit (New England Biolabs) according to the manufacturer's instructions. Real-time PCR was done with the Power SYBR Green PCR master mix (Applied Biosystems). To quantify gene expression, the 2-ΔΔCt method was used. Data were normalized to an endogenous control β-actin, and all experiments were performed in triplicate. Primer sequences are listed as follows:

 Human β-actin-Forward: 5′- AGGCACCAGGGCGTGAT -3′
 Human β-actin-Reverse: 5′- CGTCACACTTCATGATGGAATTGA -3′
 Human PDK1-Forward: 5′- CGGATCAGAAACCGACACA -3′
 Human PDK1-Reverse: 5′- ACTGAACATTCTGGCTGGTGA -3′
 Human SREBP-1c Forward: 5′- GCGCCTTGACAGGTGAAGTC -3′
 Human SREBP-1c Reverse: 5′- GCCAGGGAAGTCACTGTCTTG -3′
 Human FAS-Forward: 5′- AGTACACACCCAAGGCCAAG -3′
 Human FAS-Reverse: 5′- GGATACTTTCCCGTCGCATA -3′
 Human RNF146-Forward: 5′- GAGAAAAGACTGCGAGGTGG -3′
 Human RNF146-Reverse: 5′- GATGCCTGCCACAAAAATAAA-3′
 Human TNKS1-Forward: 5′- GACCCAAACATTCGGAACAC-3′
 Human TNKS1-Reverse: 5′- GCAGCTTCTAGGAGTTCGTCTT-3′
 Human TNKS2-Forward: 5′- ACGTGGAACGAGTCAAGAGG-3′
 Human TNKS2-Reverse: 5′- TTGCACCATTCTGAAGCAAA-3′
 Mouse β-actin-Forward 5′- CGGTTCCGATGCCCTGAGGCTCTT -3′
 Mouse β-actin-Reverse 5′- CGTCACACTTCATGATGGAATTGA -3′
 Mouse PDK1-Forward 5′- ACAAGGAGAGCTTCGGGGTGGATC -3′
 Mouse PDK1-Reverse 5′- CCACGTCGCAGTTTGATTTATGC -3′
 Mouse FAS-Forward: 5′- GCTGCGGAAACTTCAGGAAAT -3′
 Mouse FAS-Reverse: 5′- AGAGACGTGTCACTCCTGGACTT -3′

**Cell culture and cell transfection.** The HEK293T, HEK293A, H1299, H2087, MCF7, H358, U2OS, SNU475, HeLa, and A549 cell lines were purchased from the American Type Culture Collection (ATCC) and cultured under conditions specified by the manufacturer. The LKR13 mouse lung cancer cells were kindly provided by Dr. Tyler Jacks (Massachusetts Institute of Technology) and maintained in RPMI 1640 supplemented with 10% fetal bovine serum and 1% penicillin and streptomycin. Plasmid transfection was done with Mirus transfection reagents according to the manufacturer's protocol.

All cell lines used were tested for mycoplasma contamination, and were authenticated by DNA fingerprints.

**Drug library screening for AMPK regulators.** We used a small-molecular set with 406 drugs including FDA-approved agents/clinical candidates/small molecular inhibitors (provided by Dr. Clifford C Stephan, Institute of Bioscience and Technology, Texas A&M University), with metformin/AICAR as positive controls for the screening. U2OS cells were seeded in 96-well plates and then were treated with 406 drugs (5 μM) and metformin (5 mM)/AICAR (5 mM) for 12 h. Cell lysates were subjected to a p-AMPK ELISA assay by using the PathScan Phospho-AMPKα (Thr172) Sandwich ELISA Kit (Cell Signaling Technology), and relative AMPK-pT172 levels were determined.

**Lipid droplet staining.** Cells were cultured on coverslips for 36 h and incubated in medium with 200 μM sodium oleate overnight and then fixed with 4% paraformaldehyde for 10 min at room temperature, followed by stained with 0.5 μM BODIPY 493/503 (Invitrogen) for 30 min, and mounted with anti-fade reagent with DAPI. Fluorescence images were acquired with a Nikon ECLIPSE E800 fluorescence microscope with a Nikon Plan Fluor 10 × objective lens. Cells with more than 20 lipid droplets (LDs) were counted as BODIPY-positive cells. Each experiment was repeated three times unless otherwise noted.

**Cell proliferation and colony-formation assays.** For assays of proliferation, equal numbers of cells (10,000 cells per dish) were plated in 6-cm dishes and treated 24 h later with 1 μM of XAV939 or G007-LK. Cells were then collected and counted 6 days later.

For assays of colony formation, equal numbers of cells were seeded in six-well plates in triplicates (500 cells per well) and treated 24 h later with 1 μM of XAV939 or G007-LK. Medium was replenished every 3 days and cells were incubated for 10 days. Colonies were stained with Coomassie blue and quantified by using a Gel Doc Imager instrument with Quantity One software (BioRad).

**Analysis of nucleotides by IC-HRMS.** To determine the ratios of AMP and ATP in cell samples, extracts were prepared and analyzed by high-resolution mass spectrometry (HRMS). Approximately 80% confluent cells were seeded in 10 cm dishes in triplicate. 293 A Cells were washed and incubated in fresh medium without or with metformin(5 mM)/XAV939(5 μM)/G007-LK(5 μM) for 12 h. Then the cells were quickly washed with ice-cold PBS to remove extra medium components. Metabolites were extracted using 0.1% ammonium hydroxide in 90/10 (v/v) acetonitrile/water. Samples were centrifuged at $17,000 \times g$ for 5 min at 4 °C, and supernatants were transferred to clean tubes, followed by evaporation to dryness under nitrogen. Samples were reconstituted in deionized water, then 5 μl was injected into a Thermo Scientific Dionex ICS-5000 + capillary ion chromatography (IC) system containing a Thermo IonPac AS11 250 × 2 mm 4 μm column. IC flow rate was 350uL/min (at 30 °C) and the gradient conditions are as follows: started with an initial 20 mM KOH, increased to 100 mM at 10 min, held 100 mM for 10 mins. The total run time is 25 min. To assist the desolvation for better sensitivity, methanol was delivered by an external pump and combined with the eluent via a low dead volume mixing tee. Data were acquired using a Thermo Orbitrap Fusion Tribrid Mass Spectrometer under ESI negative mode. Then the raw files were imported to Thermo Trace Finder software for final analysis.

**In vitro PARylation assay.** Samples containing 0.2 μg recombinant baculovirus-derived TNKS1/2 (purchased from Sigma–Aldrich, SPR0422 and SPR0424) and 2 μg recombinant E. coli-derived GST-LKB1 (or SFB-LKB1 proteins purified from HEK293T cells) were incubated in 50 μL PARP reaction buffer (50 mM Tris-HCl, pH 8.0, 4 mM $MgCl_2$, 0.2 mM dithiothreitol) with or without 25 μM NAD+ (New

England Biolabs) at 25 °C for 30 min. Reactions were terminated by the addition of 2 × sample buffer and the samples subjected to western blotting.

**Mass spectrometry analysis for identifying ADP-ribosylation sites**. 12 µg recombinant E. coli-derived GST-LKB1 were incubated without or with 1.5 µg recombinant baculovirus-derived TNKS1 (purchased from Sigma–Aldrich, SPR0422) in 50 µL PARP reaction buffer (50 mM Tris-HCl, pH 8.0, 4 mM $MgCl_2$, 0.2 mM dithiothreitol) with 25 µM $NAD^+$ (New England Biolabs) at 25 °C for 30 min. Reactions were terminated by frozen at −80 °C for further preparation for mass spectrometry analysis.

Samples were digested with trypsin at a 1:100 (enzyme/substrate) ratio for 2 h at room temperature. m-aminophenylboronic acid-agarose (Sigma) beads were washed three times using the buffer (1% SDS, 100 mM HEPES (PH8.5)). The samples were incubated with these beads for 1 h at RT and then washed with the SDS wash buffer (1% SDS, 100 mM HEPES (PH8.5), 150 mM NaCl) for five times. Peptides were eluted by incubating the beads with 0.5 M NH2OH overnight at RT on an end-to-end rotator. Released peptides were desalted on SepPak C18 columns according to the manufacturer's instructions.

Samples were analyzed by LC-MS/MS on an LTQ-Velos Pro Orbitrap mass spectrometer (Thermo Fischer Scientific) using a top20 method. The isolation window and the minimal signal threshold for MS/MS experiments were set to be 2 Th and 500 counts, respectively. The ReAdW.exe program was used to convert the raw files into the mzXML format (http://sashimi.svn.sourceforge.net/viewvc/sashimi/). MS/MS spectra were searched against a composite database of the human IPI protein database and its reversed complement using the Sequest algorithm. Search parameters allowed for a static modification of 57.02146 Da on cysteine and a dynamic modification of addition of 15.0109 Da to aspartic acid and glutamic acid, the stable isotope (10.00827 Da and 8.01420 Da) on arginine and lysine, respectively). Search results were filtered to include <1% matches to the reverse database by the linear discriminator function using parameters including Xcorr, dCN, missed cleavage, charge state (exclude 1 + peptides), mass accuracy, peptide length and fraction of ions matched to MS/MS spectra. Localization of ADP-ribosylation sites was assessed by the ModScore algorithm based on the observation of site-specific fragment ions. Sites with scores $\geq 13$ ($P \leq 0.05$) were considered localized. Peptide quantification was performed as previously described. ADP-ribosylation motifs were extracted using the Motif-X algorithm with a significance threshold of $P < 0.001$.

**In vivo ubiquitination assay**. HEK293T cells were transfected with HA-ub (or mutants of HA-ub) and the indicated plasmids for 24 h (or 293 A cells were transduced with the indicated shRNA), after which the cells were treated with 10 µM MG132 for 6 h, harvested, and lysates were subjected to immunoprecipitation with the indicated antibodies. Ubiquitination of LKB1 was detected by western blotting.

**Measurement of hepatic triglyceride and cholesterol levels**. Lipids were extracted from mouse livers by using the Lipids Extraction Kit (BioVision K216-50) according to the manufacturer's instructions. Hepatic triglyceride and cholesterol levels in the extracted lipids were measured enzymatically with a Triglyceride Colorimetric Assay Kit (Cayman Chemical 100100303) and a Cholesterol Fluorometric Assay Kit (Cayman Chemical 100107640).

**Lactate production assay**. Cells were seeded in six-well plates and cultured overnight, and the medium was changed to no-serum DMEM for 12 h. The medium was then removed and lactate concentration was determined by using lactate test strips with a Lactate Plus meter (Nova Biomedical). The remaining cells were harvested and counted, and lactate production was calculated as lactate concentration per $10^4$ cells.

**Cell viability assay**. Cells seeded at low density in 96-well plates in triplicate overnight and then subjected to either glucose starvation for 24 h, or 8 mM metformin for 3 days. Cell death was determined by MTT assay with a Cell Growth Determination Kit (Sigma–Aldrich CGD1-1KT) according to the manufacturer's instructions. Cell death was presented as a percentage of the untreated controls.

**In vivo xenograft study**. All animal experiments were performed in accordance with a protocol approved by the Institutional Animal Care and Use Committee (IACUC) of MD Anderson Cancer Center. Our power calculations indicated that use of 10 mice per treatment group could identify the expected effect of G007-LK on tumor weight with 100% power. For the G007-LK-treated LKR13 tumor model, we injected tumor cells ($0.5 \times 10^6$ LKR13 cells or $0.25 \times 10^6$ LKR13-LKB1-knockout cells in 100 µL of PBS) into 8-week-old female 129 S mice. Tumor size was measured as indicated using a caliper, and tumor volume was calculated using the formula $0.5 \times L \times W^2$. When tumor volume of each groups reached a calculated average of 100 mm³, and the animals in each group were randomized into two subgroups (10 mice per group): (1) vehicle control and (2) G007-LK. Mice were treated for 4 weeks, and G007-LK was dosed (30 mg/kg i.p. injection. G007-LK powder was dissolved to a final concentration of 10 mg/ml in 20% DMSO, 20% Cremophor EL, 10% ethanol, 50% PBS) once daily for 5 days/week. For the G007-LK-treated LKR13-inducible-TNKS1 tumor model, we injected tumor cells ($0.5 \times$

$10^6$ LKR13 cells in 100 µL of PBS) into 8-week-old female 129 S mice. Tumor size was measured as indicated until tumor volume of each groups reached a calculated average of 100 mm³, and the animals in each group were randomized into two subgroups (eight mice per group): (1) vehicle control and (2) G007-LK and treated as previously. For the G007-LK-treated HeLa tumor model, we injected tumor cells ($2 \times 10^6$ in 100 µL of PBS) into to 6-week-old female BALB/c nude mice. After 4 weeks, the mice in each group were randomized into two subgroups (six mice per group): (1) vehicle control and (2) G007-LK. Mice were treated for 4 weeks same as previous. Mice were euthanized and tumors were removed, photographed, and weighed at 4 weeks after treatment.

For the *db/db* mouse experiment, ten-week-old male C57BL/KsJ-Leprdb/Leprdb (*db/db*) mice were randomly assigned to 4 groups (five mice per group) and a control group (3 C57 control mice). One group was given G007-LK (30 mg/kg by i.p. injection) once daily for 5 days/week, and the other groups were injected with metformin (250 mg per kg mouse body weight dissolved in PBS, by i.p. injection, once daily), the combination of G007-LK and metformin, or vehicle only. Mice were fasted for 12 h before blood glucose was measured. Blood glucose levels were measured every 5 days by using an OneTouch Ultra glucometer (LifeScan). Food intake, water consumption, and urine output of each group were measured after treatment for 3 weeks. All four groups of mice were treated for 30 days. Finally, for plasma insulin measurements, on day 30 the mice were fasted for 12 h and blood was collected from the retro-orbital sinus; plasma was obtained after centrifugation at $3000 \times g$ and plasma insulin levels were estimated with a mouse insulin ELISA kit (Millipore). Mice were then killed and their livers extracted for real-time PCR, lipid extraction, and western blot analyses.

**Crystal structure assessment for LKB1 complex and protein–protein docking**. The crystal structures for LKB1 (PDB ID: **2WTK**)[13], the TNKS Ank repeat 2 (PDB ID: **5JHQ**)[49] were taken from the Research Collaboratory for Structural Bioinformatics (RCSB) Protein Data Bank. Pre-processing was done in the Chemical Computing Group Molecular Operating Environment before docking and consisted of structure preparation, protonation, assignment of partial charges, and energy refinement. Protein–protein docking was performed on the ZDOCK Server[50].

Six different dockings covering the two experimentally determined binding regions on LKB1 (42–49, 86–93) were performed to determine possible binding poses for LKB1–TNKS Ank repeat two interactions. The following residues were pre-selected as contact points on LKB1: R86, G91, R86 and G91, R86–G91, G47–Y49, and none (blind docking). No residues were pre-selected as contact points on the TNKS Ank repeat 2. Each docking produced 10 possible poses. For each binding region, the top-scoring pose with predicted interactions involving at least one of the selected residues was chosen for final output. Inter-residue interactions were detected and visualized in Schrödinger PyMOL.

Three different dockings covering the two experimentally determined ribosylation regions on LKB1 (E138 and E130) were performed to determine possible binding poses for LKB1–TNKS PARP domain interactions. The following residues were pre-selected as contact points on LKB1: E138, E130, and none (blind docking.) No residues were pre-selected as contact points on the TNKS Ank repeat 2. Each docking produced 10 possible poses. For each ribosylation region, the top-scoring pose with predicted interactions involving at least one of the selected residues was chosen for final output. Inter-residue interactions were detected and visualized in Schrödinger PyMOL.

**Human tissue IHC analysis**. All the human lung carcinoma tissue microarrays were purchased from US Biomax. While LC20812 TMA contains 126 cases of squamous cell carcinoma, 12 adeno-squamous carcinoma, 54 adenocarcinoma, and 16 normal tissues (some slices missing three cases of squamous cell carcinoma, so we have 205 cases in total for analysis). The HLug-Squ150Sur-02 TMA has 75 cases of squamous cell carcinoma with matched normal adjacent tissue with survival data. The HlugA180Su05 TMA contains 86 cases of squamous cell carcinoma with matched normal adjacent tissue and eight cases have tumor only, survival information is provided (some slices missing two cases of adenocarcinoma, so we have 91 cases in total for analysis). Samples were de-paraffinized and rehydrated. Antigen retrieval was done using 0.01 M sodium-citrate buffer (pH 6.0) in a microwave oven. To block endogenous peroxidase activity, the sections were treated with 1% hydrogen peroxide in methanol for 30 min. After 1 h pre-incubation in 10% normal goat serum to prevent nonspecific staining, the samples were incubated with anti-LKB1 (1:100. Cell Signaling Technology, 13031 T), anti-p-AMPK (1:100. Cell Signaling Technology, 25535 S), anti-TNKS1 (1:100. Abcam, ab13587) or anti-TNKS2 (1:100. raised by ourselves) at 4 °C overnight. The sections were then incubated with a biotinylated secondary antibody (Vector Laboratories, PK-6101, 1:200) and then incubated with avidin-biotin peroxidase complex solution (1:100) for 30 min at room temperature. Color was developed with the 3-amino-9-ethylcarbazole (AEC) solution. Counterstaining was carried out using Mayer's haematoxylin. All immunostained slides were scanned on the Automated Cellular Image System III (ACIS III, Dako, Denmark) for quantification by digital image analysis. A total score of protein expression was calculated from both the percentage of immunopositive cells and the immunostaining intensity. High and low protein expressions were defined using the mean score of all samples as a cutoff point.

For survival analysis, the expression of TNKS1 and p-AMPK was treated as a binary variant and divided into "high" and "low" levels. The median expression level (50th percentile) was used as the cutoff. Kaplan–Meier survival curved were compared using the log rank test with GraphPad Prism software, A P value less than 0.05 was considered as statistically significant.

**Statistical analysis.** All experiments were repeated by at least three times. We did not exclude any samples or animals from the analysis, and we used random assignment of samples or animals to different groups. Data were analyzed by the One-way ANOVA test and Pearson Chi-square analysis. We used an F-test to compare variances, a P-value of <0.05 was considered to indicate statistically significant differences.

**Reporting summary.** Further information on research design is available in the Nature Research Reporting Summary linked to this article.

## Data availability

All the data supporting the findings of this study are available within the article and its supplementary information files and from the corresponding author upon reasonable request. A reporting summary for this article is available as a Supplementary Information file.

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

## Acknowledgements

We thank our colleagues in S.L.'s laboratory for insightful discussions and assistance. We thank Christine F. Wogan at the Department of Radiation Oncology of MD Anderson Cancer Center for her expert assistance with manuscript editing. We also thank for the help from Metabolomics Facility of MD Anderson Cancer Center (funded by NIH #P30CA016672, #1S10OD012304-1 and CPRIT #RP130397). This work was supported in part by Mabuchi Program, Cancer Center Support (Core) Grant CA016672 from the National Cancer Institute, National Institutes of Health to The University of Texas MD Anderson Cancer Center. This work was also supported in part by MD Anderson Start-up funds and an Era of Hope Scholar Research award (W81XWH-09-1-0409) to J.C., and by grants from the Welch Foundation (I-1800 to Y.Y.) and NIH (GM114160 and GM122932 to Y.Y.)

## Author contributions

N.L., J.C., and S.L. conceived the project, designed the experiments. N.L. performed the experiments with assistance from Y.W., S.N., Y.Z., J.F., Y.Q., X.L., Z.C., C.S., W.D., R.Y., W.J., S.Z., Y.Y., M.H., J.C and S.L. N.L. and S.L. wrote the manuscript.

## Competing interests

The authors declare no competing interests.
