## [Peer Review File · Nature Communications]

Reviewers' comments:

Reviewer #1 : Cancer metabolism
(Remarks to the Author):

In this study, Li et al reveal that tankyrase is a negative regulator restricting LKB1 mediated AMPK activation. They show that tankyrases interacts with and ribosylates LKB1 leading to its K63-linked ubiquitination by RNF146 and attenuating LKB1/STRAD/MO25 complex formation and LKB1 activation. Notably, inactivation of tankyrase by its inhibitors enhances LKB1 mediated AMPK activation and suppresses tumorigenesis. Using diabetic mouse model, they find tankyrase inhibitor G007-LK effectively regulates liver metabolism and glycemic control in a LKB1-dependent manner. Further, tankyrase levels are found negatively correlated with p-AMPK levels and its overexpression predicts poor survival outcome of lung cancer patients. They conclude that tankyrase is a major up-stream regulator of LKB1-AMPK pathway. Overall, the manuscript is clearly written with the conclusions that are mostly supported by experimental data. The findings are potentially interesting and may provide a new therapeutic strategy for targeting human cancers and metabolic diseases. However, there are several issues remained to be addressed before it can be considered for publication in Nature Communications.

1. Auto-PARsylation of TNKS has been shown to promote its own degradation through the ubiquitin-proteasome pathway. The authors should provide the explanation of how the induction of TNKS expression level was induced by tankyrase inhibitor treatment.

2. In Figure 1e, the authors should include the blot for TNKS2 to demonstrate the endogenous interaction between TNKS2 and LKB1. Moreover, the authors should provide the data to show that TNKS2 could induce ribosylation on LKB1 at least in one panel.

3. For Figure 1f, the authors should include tankyrase inhibitor and TNKS knockdown to attest that ribosylation signal on LKB1 is specifically driven by tankyrase, but not by other PARP family members. For extended data fig 2i, the approach by immunoprecipitation with LKB1 followed by blotting with PAR cannot exclude the possibility that the PAR signal presented is from other LKB1 interacting proteins, despite the authors include tankyrase inhibitor and TNKS knockdown.

4. For the blots in ribosylation and ubiquitination assays, the authors need to include molecular weight marker to clearly reflect the expected signals for the proteins of interest. For example, the GST-LKB1 band is above TNKS1 band in extended data fig 2g, whereas the SFB-TNKS1 band is much higher than GST-LKB1 band in extended data fig 2h. The size of SFB tag cannot explain the big shift of the TNKS1 band.

5. For the immunofluorescence results such as those in Figure 3d and extended data fig 1h, 3c, it may be difficult to draw the precise conclusion due to their poor resolution. The authors should provide enlarged images with higher resolution.

6. For Figure 2e, f, g, it will be nice if the authors can include LKB1-R42/G47/R86/G91A mutant to further support the importance of ribosylation on LKB1 mediated AMPK activation.

7. In Figure 3j, the authors used the immunoprecipitation with LKB1 followed by blotting with K63-ub to demonstrate that K63-linked ubiquitination of LKB1 is induced by RNF146. However, this method may not exclude the possibility that the ubiquitination signal may derive from other LKB1 interacting proteins such as AMPK. For this reason, the authors should provide the alternative approach to support this notion.

8. In Figure 3, the authors proposed that LKB1 ribosylation facilitates its ubiquitination driven by RNF146. However, the link between ribosylation driven by tankyrases and ubiquitination driven by RNF146 was not rigorous, although the authors include one ribosylation dead mutant of LKB1 in

vitro assay. To strengthen the conclusion, the authors should manipulate tankyrases through overexpression TNSK1 alone and/or combined with the inhibitor treatment to monitor the correlation between ribosylation and ubiquitination on WT LKB1 as well as E130/138A LKB1 in vivo? And determine whether such correlation is affected upon RNF146 knockdown?

9. In Figure 4e and 4f, the authors claimed that LKB1-E130/138A mutant formed a stronger complex with STRAD and MO25. However, this LKB1 mutant formed a comparable complex with STRAD and MO25 as WT LKB1 in Figure 4e without G007-LK treatment, which is not consistent with the data shown in Figure 4f in which LKB1 mutant formed a stronger complex without RNF146 overexpression. The authors should repeat the experiments and present consistent results.

10. In Figure 5g, 5h, 5i, the authors showed increased LKB1 and AMPK phosphorylation and downregulated expression of AMPK target genes in the livers of G007-LK treated db/db mice. The authors should include c57 WT mice group in these panels to show the alteration of LKB1-AMPK cascade between c57 WT and db/db mice, which potentially contributes to diabetic related phenotype. Thus, the application of G007-LK or metformin to manipulate LKB1-AMPK cascade will be more meaningful.

11. In Figure 6a-6e, the authors showed that TNSK1 overexpression promotes tumorigenesis, accompanied by AMPK inactivation. The previous study by Huang et al (Nature volume461, pages614–620) reveals that Tankyrase inhibition stabilizes axin to antagonize Wnt signaling, which is critical for tumorigenesis. The authors should provide the evidence that the role of TNSK1 in tumorigenesis is partly through regulating LKB1-AMPK cascade.

12. AMPK is physiologically activated under energy stress (glucose deprivation) or hypoxia conditions. Can the authors address whether tankyrase-RNF146 axis is also involved in the regulation of LKB1-AMPK cascade upon energy stress and hypoxia condition? This will enhance the physiological relevance of the whole study?

13 It will be great if the authors could present a model for the study.

Reviewer #2: AMPK pathway
(Remarks to the Author):

Tankyrase inhibits LKB1/AMPK to disrupt metabolic homeostasis and promote tumorigenesis

The authors present evidence for the regulation of LKB1 by ribosylation and ubiquitination that blocks the assembly of the heterotrimer complex between LKB1, STRADa and MO25a. There is such a large body of work in the manuscript that the text often does not do justice to the data and comes over as a series of one liner statements. Approximately a third of the data presented in Figs 1-6 could be placed in the extended data to allow more text.

1 Page 1. The title seems backwards suggest "Tankyrase disrupts metabolic homeostasis and promotes tumorigenesis by inhibiting LKB1-AMPK signalling". Similarly, the running title could be "Tankyrases ribosylates LKB1 and inhibits heterotrimer assembly"

2 Page 2 line 31 interacts and ribosylates LKB1 promoting". Line 37 "suggesting that tankyrase and RNF146 are major up-stream negative regulators of the LKB1-AMPK pathway and provide a new focus for cancer and metabolic disease therapies".

3 Page 3 line 68 typo

4 Page 4 line 85 use pT172-AMPK as AMPK can be phosphorylated on multiple sites. Line 102 It would be important to point out to the reader that the drugs led to AMPK activation without any changes in LKB1 levels.

5 Page 5 line 106 and 108 need to specify in the text "had no effect in HEK293A cells". With inhibitor studies there is always a concern about off target effects. For example, if XAV939 caused dose dependent increases in cellular AMP levels this would activate AMPK independently of effects of the inhibitors on LKB1 ribosylation. This is especially important as the cells are exposed to the inhibitors for 12 hours. Data showing the effect of the inhibitors on AMP/ATP ratios (determined by mass spectrometry) is essential. Line 117 suggest "and to a lesser extent Axin stability".

6 Page 6 line 150 Fig 1F The text "Indeed, LKB1 ribosylation was observed in vivo" is insufficient to describe the panel 1F. The upper blot purports to show that IPing with the PAR antibody pulls down TNKS1 but the reader will have no confidence in the blot "blob". Similarly, the data shown in the lower section of the panel there is another "blob" for LKB1 and an apparent band at a lower size. Why are there two bands in the LKB1 input?

7 Page 7 line 158 Extended Data Fig 2j & k. There no mass spectrometry methods provided nor how the ribosylation was done or the reaction details. The Figure legends are not adequate. It would be valuable for the authors to run a simple Q-TOF experiment on LKB1 ± tankyrase treatment. This would provide the reader with a clear picture of the complexity of the LKB1 species before and after ribosylation. It is not clear why LKB1 is already ribosylated prior to treatment with TNKS1. Line 172 Fig 2a, 2b. Why is TNKS1 elevated in the G007-LK treatments? The pACC blots do not come from the same gels as the ACC total blots.

8 Page 8 line 187 Extended Data Fig 3d. The data shown do not support the claim that the E130/E138A mutant shows a greater response to metformin-induced AMPK phosphorylation. Line 192 the authors present data showing that G007-LK had a suppressive effect on LKR13 lung adenocarcinoma tumours. In Fig 2k there is substantial induction of TNKS1 in response to G007-LK treatment, why is this? Interestingly AMPK Is Required to Support Tumor Growth in Murine Kras-Dependent Lung Cancer Models according to Eichner et al Cell Metabolism. TNKS inhibitors would be expected to promote tumour growth in this case. Line 203 Fig 3A typically the pACC signal is more sensitive to treatments than the pT172 AMPK signal yet here with loss of RNF-146 it is the reverse. Enhanced phosphorylation of ACC would be consistent with suppressing fatty acid synthesis and promoting fat oxidation. It would help the reader if an explanation for the increased lactate production was provided. In Fig 3e loss of the R-146 WWE and RING domains leads to increased AMPK pT172 but why is this dramatically enhanced by the drug XAV939? This would be expected for the WT R146 but not the WWE and RIMG domain deleted constructs. The response could be consistent with the drug raising AMP levels and promoting AMPK phosphorylation.

9 Page 10 Line 246 Fig 4 While the data shown appear generally supportive of the authors' claims. There is no statistical evaluation or information on the number of times the experiments have been independently replicated.

10 Figure 5 Panel g shows the strong induction of TNKS1 & 2 in the presence of the inhibitor G007-LK. This raises a question about the therapeutic use of these inhibitors and whether the responses would become refractory with time.

Overall this is an exciting piece of work but requires revision to make it digestible to the broader reader. Key additional experiments include testing whether the drugs alter the cell adenylate charge ie increase the AMP/ATP ratio. It would greatly help to see Q-TOF data of intact LKB1 before and after ribosylation.

Point-to-point Response:

Reviewer #1: Cancer metabolism
(Remarks to the Author):

In this study, Li et al reveal that tankyrase is a negative regulator restricting LKB1 mediated AMPK activation. They show that tankyrases interacts with and ribosylates LKB1 leading to its K63-linked ubiquitination by RNF146 and attenuating LKB1/STRAD/MO25 complex formation and LKB1 activation. Notably, inactivation of tankyrase by its inhibitors enhances LKB1 mediated AMPK activation and suppresses tumorigenesis. Using diabetic mouse model, they find tankyrase inhibitor G007-LK effectively regulates liver metabolism and glycemic control in a LKB1-dependent manner. Further, tankyrase levels are found negatively correlated with p-AMPK levels and its overexpression predicts poor survival outcome of lung cancer patients. They conclude that tankyrase is a major up-stream regulator of LKB1-AMPK pathway. Overall, the manuscript is clearly written with the conclusions that are mostly supported by experimental data. The findings are potentially interesting and may provide a new therapeutic strategy for targeting human cancers and metabolic diseases. However, there are several issues remained to be addressed before it can be considered for publication in Nature Communications.

Thanks for the nice summary of our manuscript and please see below for additional information.

1. Auto-PARsylation of TNKS has been shown to promote its own degradation through the ubiquitin-proteasome pathway. The authors should provide the explanation of how the induction of TNKS expression level was induced by tankyrase inhibitor treatment.

The reviewer is correct in pointing out that “Auto-PARsylation of TNKS has been shown to promote its own degradation through the ubiquitin-proteasome pathway”. Thus, tankyrase inhibitor treatment would suppress auto-PARsylation of TNKS, and therefore block its degradation via proteasome pathway, which lead to TNKS protein stabilization. This is the reason that tankyrase inhibitor treatment induces TNKS protein level.

2. In Figure 1e, the authors should include the blot for TNKS2 to demonstrate the endogenous interaction between TNKS2 and LKB1. Moreover, the authors should provide the data to show that TNKS2 could induce ribosylation on LKB1 at least in one panel.

We appreciate these suggestions! We conducted another set of experiment and included the blot for TNKS2 to show the endogenous interaction between TNKS2 and LKB1 in the revised manuscript (please see revised **Figure 1e**). We also showed that TNKS2 could ribosylate LKB1 (please see revised **Figure 1g**).

3. For Figure 1f, the authors should include tankyrase inhibitor and TNKS knockdown to attest that ribosylation signal on LKB1 is specifically driven by tankyrase, but not by other PARP family members. For extended data fig 2i, the approach by immunoprecipitation with LKB1 followed by blotting with PAR cannot exclude the possibility that the PAR signal presented is from other LKB1 interacting proteins, despite the authors include tankyrase inhibitor and TNKS knockdown.

We agree with the review that from **Fig 1f** we cannot conclude that ribosylation signal on LKB1 is specifically driven by tankyrase, which is the reason that we stated in page 6 of the manuscript that “LKB1 ribosylation was observed in vivo (Fig. 1f)”. Similarly from the data presented in Supplementary

Fig. 2i, we could not exclude the possibility that the PAR signal was from other LKB1 interacting proteins. To address these concerns, we performed the in vitro ribosylation assay using *E.coli* purified TNKS1/2 and LKB1 to show that TNKS1/2 can directly ribosylate LKB1 (please see revised **Figure 1g**). Our in vitro ribosylation assay performed in the presence of tankyrase inhibitors XAV939 and G007-LK (please see revised Supplementary **Fig. 2g**) indicated that tankyrase-induced LKB1 ribosylation was abolished by tankyrase inhibitors. Together, these data suggested that LKB1 can be directly ribosylated by TNKS1/2.

Following the reviewer's suggestion, we want to further determine that the ribosylation signal on LKB1 was driven by tankyrase, but not other PARP family members. Thus, we examined the ribosylation signal on LKB1 without or with treatment of tankyrase and PARP1/2 inhibitors (please see Supplementary **Fig. 2j**). We showed that tankyrase inhibitors XAV939 and G007-LK could block the ribosylation signal of LKB1 but not that of PARP1, and PARP1/2 inhibitors olaparib and PJ34 could block PARP1 but not LKB1 ribosylation. These data indicate that the ribosylation signal on LKB1 was driven by tankyrase but likely not by other PARP family members, at least not by PARP1/2.

4. For the blots in ribosylation and ubiquitination assays, the authors need to include molecular weight marker to clearly reflect the expected signals for the proteins of interest. For example, the GST-LKB1 band is above TNKS1 band in extended data fig 2g, whereas the SFB-TNKS1 band is much higher than GST-LKB1 band in extended data fig 2h. The size of SFB tag cannot explain the big shift of the TNKS1 band.

Thanks for the suggestions! We have included molecular weight marker for the blot in ribosylation and ubiquitination assays (please see revised **Figures**). The GST-LKB1 band is about 75-80 kDa. The TNKS1 we used in extended Figure 2g and in Figure 1g was enzymatic active form of TNKS1 (amino acids 1000-1328 with GST-tag, molecular weight is about 60-62 kDa, purchased from Sigma, catalog# SPR0422, please see details in **Methods** of the manuscript), so that the GST-LKB1 band is above TNKS1 band in extended figure 2g and Figure 1g. In Supplementary **Fig. 2h**, we used SFB-TNKS1 (full-length of TNKS1, expressed and purified from 293T cells) with molecular weight of 150 kDa, so that the SFB-TNKS1 band is much higher than GST-LKB1 band in this figure. We have now provided these details in the revised figure legends.

5. For the immunofluorescence results such as those in Figure 3d and extended data fig 1h, 3c, it may be difficult to draw the precise conclusion due to their poor resolution. The authors should provide enlarged images with higher resolution.

Following the suggestion of the reviewer, we provided enlarged images with higher resolution for Figure 3d and extended data figure 1h, 3c (please see revised Supplementary **Fig. 1i, Fig 3c, Fig 4d**).

6. For Figure 2e, f, g, it will be nice if the authors can include LKB1-R42/G47/R86/G91A mutant to further support the importance of ribosylation on LKB1 mediated AMPK activation.

Thanks for your suggestion. We performed the experiments in Figure 2e, f, g including LKB1-R42/G47/R86/G91A mutant and showed that this mutant exhibited similar phenotypes as LKB1-E130/138A mutant (in comparison with wildtype-LKB1), suggesting that both the tankyrase-binding activity of LKB1 and the ribosylation of LKB1 by tankyrase are important for LKB1 mediated AMPK activation (please see revised **Figure 2e, f, g**).

7. In Figure 3j, the authors used the immunoprecipitation with LKB1 followed by blotting with K63-ub to

demonstrate that K63-linked ubiquitination of LKB1 is induced by RNF146. However, this method may not exclude the possibility that the ubiquitination signal may derive from other LKB1 interacting proteins such as AMPK. For this reason, the authors should provide the alternative approach to support this notion.

We agree with this reviewer that the method in Figure 3j could not exclude the possibility that ubiquitination signal may derive from other LKB1 interacting proteins. As suggested, we performed ubiquitination assay under denature immunoprecipitation condition using k63-ub antibody. We used ubiquitin antibody to detect total k63-ubiquitin and LKB1 antibody to detect k63-chain ubiquitinated LKB1. The denature IP assay helped to exclude LKB1 binding proteins. Under this condition, we obtained the same results as we previously presented (please see revised **Figure 3h**), indicating that k63-linked LKB1 ubiquitination is induced by RNF146.

8. In Figure 3, the authors proposed that LKB1 ribosylation facilitates its ubiquitination driven by RNF146. However, the link between ribosylation driven by tankyrases and ubiquitination driven by RNF146 was not rigorous, although the authors include one ribosylation dead mutant of LKB1 in in vitro assay. To strengthen the conclusion, the authors should manipulate tankyrases through overexpression TNSK1 alone and/or combined with the inhibitor treatment to monitor the correlation between ribosylation and ubiquitination on WT LKB1 as well as E130/138A LKB1 in vivo? And determine whether such correlation is affected upon RNF146 knockdown?

As suggested, we compared the ubiquitination of wildtype-LKB1 and LKB1-E130/138A mutant. We also monitored the ubiquitination status of wildtype-LKB1 vs LKB1-E130/138A under G007-LK treatment or RNF146 knockdown condition (please see revised Supplementary **Fig. 4h**). Data from Supplementary **Fig. 4h** showed that tankyrase inhibitor G007-LK suppressed ribosylation and the subsequent ubiquitination of wildtype-LKB1, RNF146 knockdown only blocked ubiquitination of wildtype-LKB1 but had no effect on ribosylation. As for the LKB1-E130/138A mutant, since this mutant is defective in ribosylation, G007-LK treatment or RNF146 knockdown could not affect the ubiquitination status of this mutant LKB1. These data indicate that tankyrase-mediated ribosylation is linked to RNF146-mediated LKB1 ubiquitination.

9. In Figure 4e and 4f, the authors claimed that LKB1-E130/138A mutant formed a stronger complex with STRAD and MO25. However, this LKB1 mutant formed a comparable complex with STRAD and MO25 as WT LKB1 in Figure 4e without G007-LK treatment, which is not consistent with the data shown in Figure 4f in which LKB1 mutant formed a stronger complex without RNF146 overexpression. The authors should repeat the experiments and present consistent results.

Thanks for pointing this out! Actually, as shown in both Figure 4e and 4f, LKB1-E130/138A mutant formed a stronger complex with STRAD and MO25 without any treatment, although the difference was not that obvious in Figure 4e. We repeated the experiments in Figure 4e and obtained similar results, i.e. LKB1-E130/138A mutant formed a stronger complex with STRAD and MO25 without G007-LK treatment (please see revised **Figure 4e**). This supports the view that ribosylation of LKB1 by TNKS destabilizes the LKB1/STRAD/MO25 complex.

10. In Figure 5g, 5h, 5i, the authors showed increased LKB1 and AMPK phosphorylation and downregulated expression of AMPK target genes in the livers of G007-LK treated db/db mice. The authors should include c57 WT mice group in these panels to show the alteration of LKB1-AMPK cascade between c57 WT and db/db mice, which potentially contributes to diabetic related phenotype. Thus, the

application of G007-LK or metformin to manipulate LKB1-AMPK cascade will be more meaningful.

Thank you for the suggestion. We included the c57 WT mice group and repeated the experiments. As shown in revised **Figure 5g/h/i**, c57 WT mice exhibited higher p-AMPK/p-LKB1 in protein level and lower PDK1/FAS in mRNA level in comparison with db/db mice. When db/db mice were treated metformin, G007-LK, or the combination, the p-AMPK and PDK1/FAS levels were similar to those in c57 WT mice (please see revised **Figure 5g/h/i**). These results are consistent with other data presented in **Figure 5**, indicating that tankyrase inhibitor G007-LK act similarly as metformin to regulate liver metabolism and glycemic control in diabetic mice.

11. In Figure 6a-6e, the authors showed that TNSK1 overexpression promotes tumorigenesis, accompanied by AMPK inactivation. The previous study by Huang et al (Nature volume461, pages614–620) reveals that Tankyrase inhibition stabilizes axin to antagonize Wnt signaling, which is critical for tumorigenesis. The authors should provide the evidence that the role of TNSK1 in tumorigenesis is partly through regulating LKB1-AMPK cascade.

We fully agree with the reviewer's suggestion. In Figure 6a-e, we showed that TNKS1 overexpression promoted tumorigenesis by inactivating AMPK. As tankyrase has several substrates involved in tumorigenesis (e.g. Axin, PTEN, AMOT), it is critical to provide evidence that the role of TNKS in tumorigenesis is at least partly through LKB1-AMPK pathway. In **Figure 2h/i/j/k**, we showed that tankyrase inhibitor G007-LK had much more suppressive effect on LKR13 tumors than on LKR13-LKB1-ko tumors. In addition, the anti-tumor effect of G007-LK was observed in Hela-wtLKB1 tumors but not in Hela or Hela-mtLKB1 (LKB1-E130/138A mutant) tumors (please see **Figure 2l**). All these data indicate that the antitumor effect of G007-LK at least partly depends on LKB1 and therefore imply that the role of TNKS in tumorigenesis is partially through regulating LKB1-AMPK cascade.

12. AMPK is physiologically activated under energy stress (glucose deprivation) or hypoxia conditions. Can the authors address whether tankyrase-RNF146 axis is also involved in the regulation of LKB1-AMPK cascade upon energy stress and hypoxia condition? This will enhance the physiological relevance of the whole study?

Thanks for the suggestion. We tested the regulation of LKB1-AMPK pathway by tankyrase-RNF146 axis under energy stress (glucose deprivation) (please see Supplementary **Fig. 3e/4b**). In Supplementary **Fig. 3e**, the LKB1-E130/138A mutant (which could not be ribosylated by tankyrase) showed a greater protective effect on glucose starvation-induced cell death, and in Supplementary **Fig. 4b**, depletion of RNF146 also showed a protective effect on glucose starvation-induced cell death. These results suggest that tankyrase-RNF146 axis is involved in the regulation of LKB1-AMPK cascade upon energy stress. Due to the lack of expertise and equipment in our lab, we have not yet addressed the role of tankyrase-RNF146 pathway in hypoxia condition-induced AMPK activation.

13 It will be great if the authors could present a model for the study.

Thanks for pointing this out! We provided a graphical abstract to present a model (as revised Supplementary **Fig. 7**) for the study.

Reviewer #2: AMPK pathway
(Remarks to the Author):

Tankyrase inhibits LKB1/AMPK to disrupt metabolic homeostasis and promote tumorigenesis

The authors present evidence for the regulation of LKB1 by ribosylation and ubiquitination that blocks the assembly of the heterotrimer complex between LKB1, STRADa and MO25a. There is such a large body of work in the manuscript that the text often does not do justice to the data and comes over as a series of one liner statements. Approximately a third of the data presented in Figs 1-6 could be placed in the extended data to allow more text.

Thanks for the nice summary of our story.

Following the reviewer's suggestion, we moved some data presented in Figs 1-6 to extended data and added more text in the revised manuscript.

1 Page 1. The title seems backwards suggest "Tankyrase disrupts metabolic homeostasis and promotes tumorigenesis by inhibiting LKB1-AMPK signalling". Similarly, the running title could be "Tankyrases ribosylates LKB1 and inhibits heterotrimer assembly"

Thanks for the suggestions. We changed the title and running title accordingly, which we agree is better.

2 Page 2 line 31 interacts and ribosylates LKB1 promoting". Line 37 "suggesting that tankyrase and RNF146 are major up-stream negative regulators of the LKB1-AMPK pathway and provide a new focus for cancer and metabolic disease therapies".

Thanks for your suggestion! Please see changes on page 2 of revised manuscript.

3 Page 3 line 68 typo

Thanks and please see changes on page 3 of revised manuscript.

4 Page 4 line 85 use pT172-AMPK as AMPK can be phosphorylated on multiple sites. Line 102 It would be important to point out to the reader that the drugs led to AMPK activation without any changes in LKB1 levels.

Thanks for the suggestions. We made these changes on page 4 of revised manuscript.

5 Page 5 line 106 and 108 need to specify in the text "had no effect in HEK293A cells". With inhibitor studies there is always a concern about off target effects. For example, if XAV939 caused dose dependent increases in cellular AMP levels this would activate AMPK independently of effects of the inhibitors on LKB1 ribosylation. This is especially important as the cells are exposed to the inhibitors for 12 hours. Data showing the effect of the inhibitors on AMP/ATP ratios (determined by mass spectrometry) is essential. Line 117 suggest "and to a lesser extent Axin stability".

Thanks for the suggestions and please see these changes on page 5 of revised manuscript. For the effect of inhibitors on AMP/ATP ratios, we performed the experiment as suggested (please see revised Supplementary Fig. 1d, and please see "Nucleotide analysis" of Methods of the revised

manuscript). Data from extended figure 1d showed that metformin treatment increased AMP/ATP ratio while tankrase inhibitor XAV939/G007-LK had a mild effect, indicating that tankrase inhibitors and metformin may regulate AMPK activation via different mechanisms.

6 Page 6 line 150 Fig 1F The text “Indeed, LKB1 ribosylation was observed in vivo” is insufficient to describe the panel 1F. The upper blot reports to show that IPing with the PAR antibody pulls down TNKS1 but the reader will have no confidence in the blot “blob”. Similarly, the data shown in the lower section of the panel there is another “blob” for LKB1 and an apparent band at a lower size. Why are there two bands in the LKB1 input?

Thanks for pointing this out! In the upper blot of Figure 1f, we used anti-PAR antibody to immunoprecipitate the PARylated proteins in cell lysates. We detected ribosylated TNKS1 as a positive control, and we also observed ribosylated LKB1 using anti-LKB1 antibody. As for the “blob” from TNKS1 blot, TNKS1 could ribosylate itself and substrates. We normally detected much stronger signal of ribosylated TNKS1 than other tankrase substrates (e.g. Axin/LKB1), and thus sometimes there was a “blob” when we used anti-TNKS antibody to detect ribosylated TNKS.

As for the lower section of Figure 1f, we performed a denaturing immunoprecipitation assay using anti-LKB1 antibody followed by blotting with anti-PAR antibody. We observed a band at the same molecular weight as LKB1 (we included molecular weight marker in the revised **Figure 1f**), indicating that LKB1 was PARylated in cells. As for the two bands in the LKB1 input, there was a small mistake as we loaded the same input twice for western blot. We already repeated this experiment and the new data was shown in revised **Figure 1f**.

7 Page 7 line 158 Extended Data Fig 2j & k. There no mass spectrometry methods provided nor how the ribosylation was done or the reaction details. The Figure legends are not adequate. It would be valuable for the authors to run a simple Q-TOF experiment on LKB1 ± tankrase treatment. This would provide the reader with a clear picture of the complexity of the LKB1 species before and after ribosylation. It is not clear why LKB1 is already ribosylated prior to treatment with TNKS1. Line 172 Fig 2a, 2b. Why is TNKS1 elevated in the G007-LK treatments? The pACC blots do not come from the same gels as the ACC total blots.

The details about the ribosylation assay was provided in “In vitro PARsylation assay” in the **Methods** of this manuscript, which was based on our previously published protocol (Zhang et al Nature Methods, 2013). These experiments were performed by our collaborator Dr. Yonghao Yu who previously developed these mass spectrometry technologies.

We had two groups of in vitro PARsylation experiments for mass spectrometry analysis, LKB1 alone and LKB1 with TNKS1 (please see Supplementary **Fig. 2k/l**). We observed several putative ribosylation sites on LKB1 prior to treatment with TNKS1, presumably as a result of basal PARylation. However, we have performed additional experiments to identify the functionally relevant modification sites (please see Supplementary **Fig. 2m/n**).

As auto-PARsylation of TNKS has been shown to promote its own degradation through the ubiquitin-proteasome pathway, tankrase inhibitor treatment could suppress auto-PARsylation of TNKS, block its proteasome degradation, and thus lead to TNKS protein stabilization. Therefore, tankrase inhibitor G007-LK simultaneously inhibits tankrase enzymatic activity and stabilizes tankrase protein.

We also performed Western blotting using protein samples from **Figure 2a/b**, and detected p-ACC and total ACC level from the same gels by striping and reprobing (please see revised **Figure 2a/b**).

8 Page 8 line 187 Extended Data Fig 3d. The data shown do not support the claim that the E130/E138A mutant shows a greater response to metformin-induced AMPK phosphorylation. Line 192 the authors present data showing that G007-LK had a suppressive effect on LKR13 lung adenocarcinoma tumours. In Fig 2k there is substantial induction of TNKS1 in response to G007-LK treatment, why is this? Interestingly AMPK Is Required to Support Tumor Growth in Murine Kras-Dependent Lung Cancer Models according to Eichner et al Cell Metabolism. TNKS inhibitors would be expected to promote tumour growth in this case. Line 203 Fig 3A typically the pACC signal is more sensitive to treatments than the pT172 AMPK signal yet here with loss of RNF-146 it is the reverse. Enhanced phosphorylation of ACC would be consistent with suppressing fatty acid synthesis and promoting fat oxidation. It would help the reader if an explanation for the increased lactate production was provided. In Fig 3e loss of the R-146 WWE and RING domains leads to increased AMPK pT172 but why is this dramatically enhanced by the drug XAV939? This would be expected for the WT R146 but not the WWE and RIMG domain deleted constructs. The response could be consistent with the drug raising AMP levels and promoting AMPK phosphorylation.

Thanks for the comments! In Supplementary **Fig. 3d**, Hela-wildtype-LKB1 cells showed significant induction of AMPK phosphorylation when treated with 2mM metformin, while Hela-LKB1-E130/138A cells showed similar activation of AMPK when treated with 1mM metformin. Thus, we claimed that “the E130/138A mutant shows a greater response to metformin-induced AMPK phosphorylation”.

As auto-PARsylation of TNKS has been shown to promote its own degradation through the ubiquitin-proteasome pathway, tankrase inhibitor treatment could suppress auto-PARsylation of TNKS, block its proteasome degradation, and lead to TNKS protein stabilization. Therefore, tankrase inhibitor G007-LK treatment led to inhibition of tankrase enzymatic activity and stabilization of tankrase protein level as shown in **Figure 2k**.

Thanks for pointing out the roles of AMPK in tumor growth. According to Eichner et al Cell Metabolism 2018, Eichner and colleagues concluded that “AMPK deletion is detrimental to the growth of $Kras^{G12D}p53^{-/-}$ (KP) NSCLC; AMPK loss does not phenocopy LKB1 loss in $Kras^{G12D}$ -dependent NSCLC”, indicating that AMPK is required to support tumor growth only in $Kras^{G12D}p53^{-/-}$ (KP) NSCLC. LKR13 cells are $Kras^{G12D}$ -mutant but p53-wildtype murine lung cancer cell line (Meylan et al Nature 2009). It is not surprising that TNKS inhibitor could promote activation of LKB1/AMPK cascade and suppress tumor growth, and as in most cases, depletion/inhibition of LKB1/AMPK would promote tumorigenesis.

We agree with this reviewer that typically the pACC signal is more sensitive to treatments than the pT172 AMPK signal, as you can see data from **Figure 1b/c**, the fold change of pACC is more dramatic than that of pT172 AMPK. However sometimes, the pACC antibody did not work as well as pAMPK antibody for Western blotting. We exposed the WB film much longer to show a clear band of pACC, which may reduce the sensitivity of pACC signal.

As AMPK/ACC cascade function to coordinate metabolism and biosynthesis (Faubert et al Cell Metabolism 2013; Deng et al Molecular Cell 2016), inhibition/inactivation of AMPK/ACC cascade led to increased glucose consumption and lactate production. Our data presented in **Figure 3a/b** and Supplementary **Fig. 4c/d** showed that depletion of RNF146 induced activation of AMPK/ACC, and the

subsequent decrease of AMPK downstream gene expression, lactate production and lipid droplet formation, which is consistent with current literature and concepts in the field.

In **Figure 3e** (revised **Figure 3c**), we showed that overexpression of wildtype-RNF146 but not WWE and RING domain deletion mutant of RNF146 led to suppression of pAMPK, indicating that the recognition of ribosylated LKB1 by WWE domain and ubiquitination of LKB1 by RING domain of RNF146 all contribute to RNF146-mediated AMPK regulation. As tankyrase inhibition led to AMPK activation (please see **Figure 1b** and Supplementary **Fig. 1c**), AMPK phosphorylation was dramatically enhanced by XAV939, and XAV939 treatment reversed the RNF146-induced AMPK suppression (please see revised **Figure 3c**). Tankyrase inhibitors promoted AMPK phosphorylation while they did not affect cellular AMP/ATP ratio (please see revised extended **Figure 1d**)

9 Page 10 Line 246 Fig 4 While the data shown appear generally supportive of the authors' claims. There is no statistical evaluation or information on the number of times the experiments have been independently replicated.

Thanks for pointing this out! In the "statistical analysis" part of **Methods** (please see page 24 of the manuscript), we stated that "All experiments were repeated by at least three times".

10 Figure 5 Panel g shows the strong induction of TNKS1 & 2 in the presence of the inhibitor G007-LK. This raises a question about the therapeutic use of these inhibitors and whether the responses would become refractory with time.

We agree with this reviewer that it is important to know the response to tankyrase inhibitor G007-LK in xenograft study. We treated the LKR13 tumors with G007-LK for 4 weeks (please see **Figure 2k**) and db/db mice for 4 weeks (please see **Figure 5g**). We observed tumor suppression, stabilization of tankyrase and activation of AMPK, indicating that G007-LK worked well during this period. We have not yet determined whether or not the responses would become refractory with time, but this is an area for investigation in the future.

Overall this is an exciting piece of work but requires revision to make it digestible to the broader reader. Key additional experiments include testing whether the drugs alter the cell adenylate charge ie increase the AMP/ATP ratio. It would greatly help to see Q-TOF data of intact LKB1 before and after ribosylation.

Reviewers' comments:

Reviewer #1 (Remarks to the Author):

The authors have adequately addressed all of my concerns. The study reveals novel mechanistic insight and is now suitable for publications at Nature Communications.

Reviewer #2 (Remarks to the Author):

Reviewer 2

1 Title revised & running title changed

2 Abstract revised

3 Typo corrected

4 pT72 specified

5 In response to Page 5 line 106 and 108 need to specify in the text "had no effect in HEK293A cells". With inhibitor studies there is always a concern about off target effects. For example, if XAV939 caused dose dependent increases in cellular AMP levels this would activate AMPK independently of effects of the inhibitors on LKB1 ribosylation. This is especially important as the cells are exposed to the inhibitors for 12 hours. Data showing the effect of the inhibitors on AMP/ATP ratios (determined by mass spectrometry) is essential. Line 117 suggest "and to a lesser extent Axin stability". The authors claim to have measured AMP/ATP ratios but if you look at the methods they have used it is a kit assay for cyclic AMP not AMP. Nucleotide analysis: The intercellular level of ATP and AMP were measured enzymatically with an ATP Detection Assay Kit (Cayman Chemical 700410) and a Cyclic AMP ELISA Kit (Cayman Chemical 581001) according to the manufacturer's instructions, and then the AMP/ATP ratio was calculated. It is difficult to accept that the authors know what their doing.

6 Fig 1 panel f and g revised

7 The authors have not provided any mass spectrometry methods relating to Supplement Fig 2 k & l nor have they run a Q-TOF experiment as recommended. Fig 2 b has been revised.

8 According to the authors methods they have measured cyclic-AMP not AMP.

9 It is not sufficient to say all experiments were repeated by at least three times. The data in Fig 4 needs to be supported by a statistical evaluation of all data.

10 It would be appropriate to raise the point of a potential for a refractory response in the Discussion. No further experiments are being requested.

The authors have not adequately responded to the requests of Reviewer 2 and publication at this point is not justified.

Point-by-point Response

Reviewer #1 (Remarks to the Author):

The authors have adequately addressed all of my concerns. The study reveals novel mechanistic insight and is now suitable for publications at Nature Communications.

Thank you for the support of our work.

Reviewer 2

- 1 Title revised & running title changed
- 2 Abstract revised
- 3 Typo corrected
- 4 pT72 specified

Thank you.

5 In response to Page 5 line 106 and 108 need to specify in the text “had no effect in HEK293A cells”. With inhibitor studies there is always a concern about off target effects. For example, if XAV939 caused dose dependent increases in cellular AMP levels this would activate AMPK independently of effects of the inhibitors on LKB1 ribosylation. This is especially important as the cells are exposed to the inhibitors for 12 hours. Data showing the effect of the inhibitors on AMP/ATP ratios (determined by mass spectrometry) is essential. Line 117 suggest “and to a lesser extent Axin stability”. The authors claim to have measured AMP/ATP ratios but if you look at the methods they have used it is a kit assay for cyclic AMP not AMP. Nucleotide analysis: The intercellular level of ATP and AMP were measured enzymatically with an ATP Detection Assay Kit (Cayman Chemical 700410) and a Cyclic AMP ELISA Kit (Cayman Chemical 581001) according to the manufacturer’s instructions, and then the AMP/ATP ratio was calculated. It is difficult to accept that the authors know what their doing.

Thank you for pointing out that showing the effect of inhibitors on AMP/ATP ratio is essential and we apologize for using the wrong method for this experiment. Following the reviewer’s suggestions, we used HRMS (high-resolution mass spectrometry) to measure AMP/ATP levels and calculated the AMP/ATP ratio (please see revised extended **Figure 1d**, and please see “Analysis of Nucleotides by IC-HRMS” of **Methods** of the revised manuscript). Data from extended figure 1d demonstrate that Metformin treatment significantly increased AMP/ATP ratio while tankyrase inhibitor XAV939/G007-LK had little to no effect, indicating tankyrase inhibitors and metformin regulate AMPK activation with different mechanisms. We have also provided the original data of HRMS results for the reviewer.

7 The author have not provided any mass spectrometry methods relating to Supplement Fig 2 k & l nor have they run a Q-TOF experiment as recommended. Fig 2 b has been revised.

Thanks for the comments, the mass spectrometry methods is now provided in “Mass spectrum analysis of identifying ADP-ribosylation sites” in the **Methods** of this manuscript.

The analyses of PARylated LKB1 by Q-TOF mass spectrometry is technically infeasible for the following reasons: (1) PARylation is a heterogeneous modification with the lengths of the PAR chains reaching 200 ADP-ribose units. As a result, in contrast to protein modifications where a single chemical entity is

attached to a modified protein (e.g., phosphorylation), a PARylated LKB1 protein does not possess a defined mass, with the entire modified population distributed into numerous low abundance species. This is very similar to other polymeric types of protein modifications, such as glycosylation and ubiquitination; (2) PARylation is a very unstable modification which decomposes during MS analyses; (3) PARylation is a highly acidic modification. Because each ADP-ribose contains two phosphates, and a PAR chain could contain up to hundreds of ADP-ribose units, the attachment of PAR chains results in the addition of a large number of negative charges to a protein, and prevents its efficient ionization during ESI-MS analyses.

To tackle these technical challenges, our collaborator, Dr. Yonghao Yu, previously developed a unique MS method for the analyses of PARylated proteins (Zhang et al Nature Methods, 2013 and Zhen et al., Cell Reports, 2017). This method is based on a unique reaction involving a chemical agent called hydroxylamine (NH₂OH). This compound attacks an ADP-ribosylated Asp/Glu residue, and in doing so, converts it into a hydroxamic acid derivative. This results in an addition of +15 Da on an Asp/Glu residue. Importantly, the hydroxamic acid is a stable, homogeneous and small mass tag that can be readily pinpointed by regular tandem MS experiments. This method was deployed in the current study to characterize the putative ADP-ribosylation sites of LKB1.

In the revised manuscript, we provided the original mass spectrum data of the two groups (please see Supplementary Data **Table. 3a** for LKB1 group, Supplementary Data **Table. 3b** for LKB1/TNKS1 group, and the comparison of two groups in Supplementary Data **Table. 3c**).

We observed several putative ribosylation sites on LKB1 after In vitro PARylation assay without TNKS1, As the LKB1 we used was recombinant E. coli-derived GST-LKB1, and a recent paper reported evidence of poly-ADP-ribosylation in bacterium (Cho et al Nature Communications, 2019. PMID: 30940816), they concluded that "Structure and biochemical evidence supporting poly ADP-ribosylation in the bacterium *Deinococcus radiodurans*". So It's possible that E. coli-derived GST-LKB1 already had basal level of PARylation after In vitro PARylation assay even without TNKS1. This is why we compared the differential modifications of LKB1 and the LKB1/TNKS1 groups to discover specific TNKS modified ADP-ribosylation sites on LKB1 (please see Supplementary **Fig. 2k/l**). Moreover, we performed additional experiments to identify the functionally relevant modification sites by TNKS1 (please see Supplementary **Fig. 2m/n**).

8 According to the authors methods they have measured cyclic-AMP not AMP.

We apologize for using the incorrect method to measure AMP. We now have used HRMS (high-resolution mass spectrometry) to measure AMP/ATP levels and calculate the AMP/ATP ratio (please see revised extended **Figure 1d**, and please see "Analysis of Nucleotides by IC-HRMS" of **Methods** of the revised manuscript, and the original data of HRMS results).

9 It is not sufficient to say all experiments were repeated by at least three times. The data in Fig 4 needs to be supported by a statistical evaluation of all data.

We thank the reviewer for the suggestions. We have provided relative ratios of densitometric analysis normalized to the housekeeping protein, and summarized the data by the ratio that are presented below each of the lanes on the western blots. The statistical evaluation of all data is summarized in Fig 4 (please see revised **Fig. 4**).

10 It would be appropriate to raise the point of a potential for a refractory response in the Discussion.

No further experiments are being requested.

We have added to the discussion for the potential of response to tankyrase inhibitor to become refractory over time, but at least for the duration of our preclinical experiment of giving daily i.p. injections, this was not apparent. Longer term studies will be needed to examine the potential for response refractoriness of tankyrase inhibitors as well as a study in the context of other bioavailable drugs such as AZ6102.

Reviewers' comments:

Reviewer #2 (Remarks to the Author):

Points raised

1 Resolved

2 Resolved

3 Resolved

4 Resolved

5 AMP/ATP ratio measurements following XAV939 & G007-LK treatment. In determining the AMP/ATP ratio in HEK 293A cells the authors report values of 0.1 for the basal level and 0.9- XAV939 & G007-LK an 0.3 for metformin treatment. These values are high. For example, in Zhang et al Nature 2017 548, 112-116. The basal values were 0.025. One explanation for the difference is that the author washed their cells in Milli-Q water which would have caused an osmotic shock. The cells should have been washed in ice-cold PBS. Due to the elevated basal AMP/ATP levels we cannot be confident that the inhibitors were not causing an increase in the AMP/ATP ratio. The reviewer has also confirmed that the basal AMP/ATP ratio for 293 cells is 0.01 and rises to 0.03 with phenformin treatment. The authors need to repeat this experiment using PBS washing and specify the concentration of the inhibitors used as well as the incubation time that was not specified in the Methods page 19 line 483.

6 ?

7 Resolved

8 Resolved (see under point 5)

9 The relative ratios have been included in Figure 4 but there is no statistical evaluation.

10 Resolved.

Point-by-point Response

Reviewer #2 (Remarks to the Author):

Points raised

1 Resolved

2 Resolved

3 Resolved

4 Resolved

Thank you.

5 AMP/ATP ratio measurements following XAV939 & G007-LK treatment. In determining the AMP/ATP ratio in HEK 293A cells the authors report values of 0.1 for the basal level and 0.9- XAV939 & G007-LK an 0.3 for metformin treatment. These values are high. For example, in Zhang et al Nature 2017 548, 112-116. The basal values were 0.025. One explanation for the difference is that the author washed their cells in Milli-Q water which would have caused an osmotic shock. The cells should have been washed in ice-cold PBS. Due to the elevated basal AMP/ATP levels we cannot be confident that the inhibitors were not causing an increase in the AMP/ATP ratio. The reviewer has also confirmed that the basal AMP/ATP ratio for 293 cells is 0.01 and rises to 0.03 with phenformin treatment. The authors need to repeat this experiment using PBS washing and specify the concentration of the inhibitors used as well as the incubation time that was not specified in the Methods page 19 line 483.

Thank you for pointing out that the AMP/ATP ratios are higher than other literature, and thanks for the astute suggestions that we should wash the cells with ice-cold PBS instead of ice-cold water to avoid an osmotic shock. We repeated the experiment according to the reviewer's suggestions and we did get the basal AMP/ATP ratio for 293A cells at around 0.02 which raised to 0.045 with metformin treatment (please see revised Supplementary **Figure 1d**, and please also see "Analysis of Nucleotides by IC-HRMS" in the **Methods** section of the revised manuscript), while tankyrase inhibitor XAV939/G007-LK had no effect on AMP/ATP ratio, indicating tankyrase inhibitors and metformin regulate AMPK activation via different mechanisms. We have also provided the original data of HRMS results for the reviewer. We have specified the inhibitors used as well as the incubation time prior to extraction in the "Analysis of Nucleotides by IC-HRMS" in the **Methods** section of the revised manuscript.

6 Resolved

7 Resolved

8 Resolved (see under point 5)

10 Resolved.

Thank you.

9 The relative ratios have been included in Figure 4 but there is no statistical evaluation.

We thank the reviewer for the suggestions. We have provided relative ratios of densitometric analysis normalized to the housekeeping protein, and provided statistical evaluation of all data in Fig 4 (please see revised **Fig. 4**, Figure legend 4 and Supplementary data **Table 4**).

REVIEWERS' COMMENTS:

Reviewer #2 (Remarks to the Author):

The authors have undertaken additional experiments and resolved the question of the AMP/ATP measurements. No further changes or questions to consider.

Point-by-point Response

REVIEWERS' COMMENTS:

Reviewer #2 (Remarks to the Author):

The authors have undertaken additional experiments and resolved the question of the AMP/ATP measurements. No further changes or questions to consider.

Thank you for the acceptance of this work.